# Modulating co-translational protein folding by rational design and ribosome engineering

Minkoo Ahn [1,6], Tomasz Włodarski[1,6], Alkistis Mitropoulou[1,6], Sammy H. S. Chan [1], Haneesh Sidhu[1],
Elena Plessa [1], Thomas A. Becker [2], Nediljko Budisa[3,4], Christopher A. Waudby [1], Roland Beckmann [2],
Anaïs M. E. Cassaignau [1], Lisa D. Cabrita [1✉] & John Christodoulou [1,5✉]

Co-translational folding is a fundamental process for the efficient biosynthesis of nascent polypeptides that emerge through the ribosome exit tunnel. To understand how this process is modulated by the shape and surface of the narrow tunnel, we have rationally engineered three exit tunnel protein loops (uL22, uL23 and uL24) of the 70S ribosome by CRISPR/Cas9 gene editing, and studied the co-translational folding of an immunoglobulin-like filamin domain (FLN5). Our thermodynamics measurements employing $^{19}F/^{15}N$/methyl-TROSY NMR spectroscopy together with cryo-EM and molecular dynamics simulations reveal how the variations in the lengths of the loops present across species exert their distinct effects on the free energy of FLN5 folding. A concerted interplay of the uL23 and uL24 loops is sufficient to alter co-translational folding energetics, which we highlight by the opposite folding outcomes resulting from their extensions. These subtle modulations occur through a combination of the steric effects relating to the shape of the tunnel, the dynamic interactions between the ribosome surface and the unfolded nascent chain, and its altered exit pathway within the vestibule. These results illustrate the role of the exit tunnel structure in co-translational folding, and provide principles for how to remodel it to elicit a desired folding outcome.

[1] Institute of Structural and Molecular Biology, University College London, Gower Street, London WC1E 6BT, UK. [2] Gene Center and Department of Biochemistry, Ludwig-Maximilians-Universität München, Feodor-Lynen-Straße 25, 81377 Munich, Germany. [3] Institute of Chemistry, Technische Universität Berlin, D-10623 Berlin, Germany. [4] Faculty of Science, University of Manitoba, R3T 2N2 Winnipeg, MD, Canada. [5] School of Crystallography, Birkbeck College, University of London, Malet Street, London WC1E 7HX, UK. [6] These authors contributed equally: Minkoo Ahn, Tomasz Włodarski, Alkistis Mitropoulou. ✉email: l.cabrita@ucl.ac.uk; j.christodoulou@ucl.ac.uk

The precise folding of polypeptides into their biologically active structures is crucial for underpinning cellular processes within all living organisms[1]. Most proteins are expected to fold concomitantly with translation on the ribosome, the extent of which is predicted to correlate with protein size and topological complexity[2,3]. The co-translational folding (CTF) of nascent chains (NCs) is influenced by several mechanisms ranging from interactions with the ribosome[4–8] to ribosome-associated auxiliary proteins such as molecular chaperones[9,10]. Understanding the role of the ribosome would allow the delineation of protein folding mechanisms as they take place within cells[11,12] and lead to the ability to manipulate such processes.

During biosynthesis, newly synthesizing polypeptides burrow through the ribosome's exit tunnel into the cellular environment. Its narrow confines (10–28 Å in diameter and 100 Å in length) limit the formation of the persistent tertiary structure of the NC, but helices can form both in the upper and narrower part of the tunnel near the peptidyl transferase centre (PTC) and in the vicinity of the wider vestibule region[13,14]. In E.coli's 70S ribosomal particle, the tunnel's unique geometry is shaped by the 23S ribosomal RNA (rRNA) and the ribosomal proteins uL4, uL22, uL23, uL24 and uL29 (Fig. 1a)[15–17]. Of these proteins, uL4, uL22, uL23 and uL24 all have extended loops (ranging from 10 to 25 residues in length) that protrude into the tunnel, and along with uL29, have surface-exposed globular regions that crown the perimeter of the exit tunnel. Within the tunnel, the loops create an irregular trajectory for the NC by partially occluding the upper (uL4 and uL22 at the constriction) and lower (uL23 and uL24) parts of the tunnel, and thus force direct contacts to be made with emerging NCs[18] (Fig. 1a). Mutational studies have identified pertinent roles for individual loops in modulating translational arrest[19,20], antibiotic resistance[20–22] and CTF[23]; however, an understanding of the interplay between these loops in how they shape the geometry of the tunnel, their impact on the dynamics and folding of nascent chains, and of the principles on how to model them to modulate CTF remains unexplored.

Here, we have applied a rational, structure-based design and used CRISPR-Cas9 gene editing to create novel ribosomal variants in E. coli, in order to explore how the tunnel's geometry can be modified to selectively modulate CTF. We investigated the CTF of an immunoglobulin (Ig)-like domain on these modified ribosomes using $^{19}$F NMR spectroscopy that uniquely permits quantification of folded and unfolded populations of NCs. By combining NMR spectroscopy with cryo-EM and molecular dynamics (MD) simulations and using a series of systematic modifications made to ribosomal protein loops in the exit tunnel (both extensions and truncations), we show how these loops modulate the folding energy landscape of NCs. In particular, through quantitating NC dynamics by NMR, we reveal that the uL23 and uL24 loops work in unison to regulate both the NC's spatial freedom and its interactions with the ribosome. Also, our cryo-EM structures of ribosome-nascent chain complexes show that the uL23 loop provides a further means of modulating folding preferences by independently sending the NC along distinct pathways within the tunnel. These findings show a subtle interplay between ribosomal proteins within the tunnel underpinning the dynamics, structural features and preferences of an elongating NC, pointing the way to harnessing the principles of rational design and ribosome engineering to manipulate CTF mechanisms.

## Results

**Structure and evolution of the tunnel-protruding loops within the exit tunnel.** Initially, the interactions formed between emerging NCs and the ribosomal proteins surrounding the exit tunnel in 70S E.coli ribosomes were analysed using NC structures. The latter were derived from 3D structural ensembles generated by all-atom MD simulations restrained with experimental data from NMR spectroscopy of SecM-arrested folding-competent FLN5[6] –, and folding-incompetent α-synuclein[8] – ribosome-nascent chain complexes (RNCs). In particular, the contacts made between these NCs and five universally conserved ribosomal proteins that construct the exit tunnel (uL4, uL22, uL23, uL24 and uL29) (Fig. 1a) were examined. In FLN5 RNCs at $L = 31$ ($L$, number of residues linking the FLN5 NC and the PTC), uL4 and uL22, which form the tunnel's major constriction, and uL23 and uL24 at the wider vestibule regions, respectively, make extensive direct contacts with NCs via their extended tunnel loops as shown in the contact map (Fig. 1b). In addition, discernible interactions of the FLN5 NC are observed with the exit's immediate surface regions, including a short loop between the two helices of uL29, and both the N-terminal helix (residues 15–25) and the C-terminal tail (residue 90–100) of uL23.

The tunnel's interior loops (from uL4, uL22, uL23 and uL24) similarly interact with the negatively charged C-terminal segment of α-synuclein, whereas the surface regions of uL23 and uL29 are comparatively visited less relative to the FLN5 RNCs (Fig. 1b). These observations suggest that the ribosomal tunnel loops directly interact with NCs irrespective of their sequence, influencing the trajectories of emerging NCs and consequently, how they sample conformational space to acquire structure.

Next, we explored the evolutionary conservation of length and sequence of these tunnel loops in an attempt to relate their variability to the complexity of an organism. Structure-based multiple sequence alignments reveal some variability in the upper tunnel where the uL22 loop has a well-conserved length across all domains of life, whereas the uL4 loop shows a marked difference between bacterial and eukaryotic ribosomes with the longer eukaryotic loop delimiting the presence of a second constriction site (Supplementary Fig. 1)[17,24]. At the lower end of the tunnel, the loop lengths of uL23 and uL24 vary significantly between bacteria, eukaryotes and archaea (Supplementary Fig. 1): bacterial ribosomes and mitochondrial ribosomes possess long tunnel loops of uL23 (18–20 residues) and uL24 (15–18 residues) relative to those observed in archaeal and eukaryotic cytosolic ribosomes (five residues for both uL23 and uL24), which translate comparatively more complex proteomes. Available chloroplastic ribosome structures share a short uL23 loop as is present in eukaryotes and archaea, but a long uL24 loop as seen in bacteria[25]. Such evolutionary variations suggest that the geometry of the exit tunnel has likely co-evolved alongside an organism's proteome[24] to facilitate co-translational processes including protein folding[26–29].

**Design and engineering of ribosomal mutants using CRISPR-Cas9.** Based on the findings from the structural and evolutionary sequence analysis, we designed a library of ribosome variants with modified tunnel loops (Fig. 1c, d) to explore their influence on CTF. Single truncation mutants removing eight residues from each of the uL4 loop ($4^{\Delta L}$) and the uL22 loop ($22^{\Delta L}$) were designed to investigate the steric effect of the narrow constriction site on NC folding. Within the wider vestibule region, the uL23 and uL24 loops were shortened to lengths consistent with those present in eukaryotic ribosomes (Supplementary Fig. 1) with 7 and 11-residue deletions made within the uL23 ($23^{\Delta L}$) and uL24 ($24^{\Delta L}$) loops, respectively. Double ($22^{\Delta L}23^{\Delta L}$ and $23^{\Delta L}24^{\Delta L}$) and triple ($22^{\Delta L}23^{\Delta L}24^{\Delta L}$) truncation mutants were also designed to explore potential synergistic effects from these shortened loops.

To further probe the shape of the ribosomal exit tunnel, single ($23^{+L}$, $24^{+L}$) and double ($23^{+L}24^{+L}$) insertion mutants of uL23

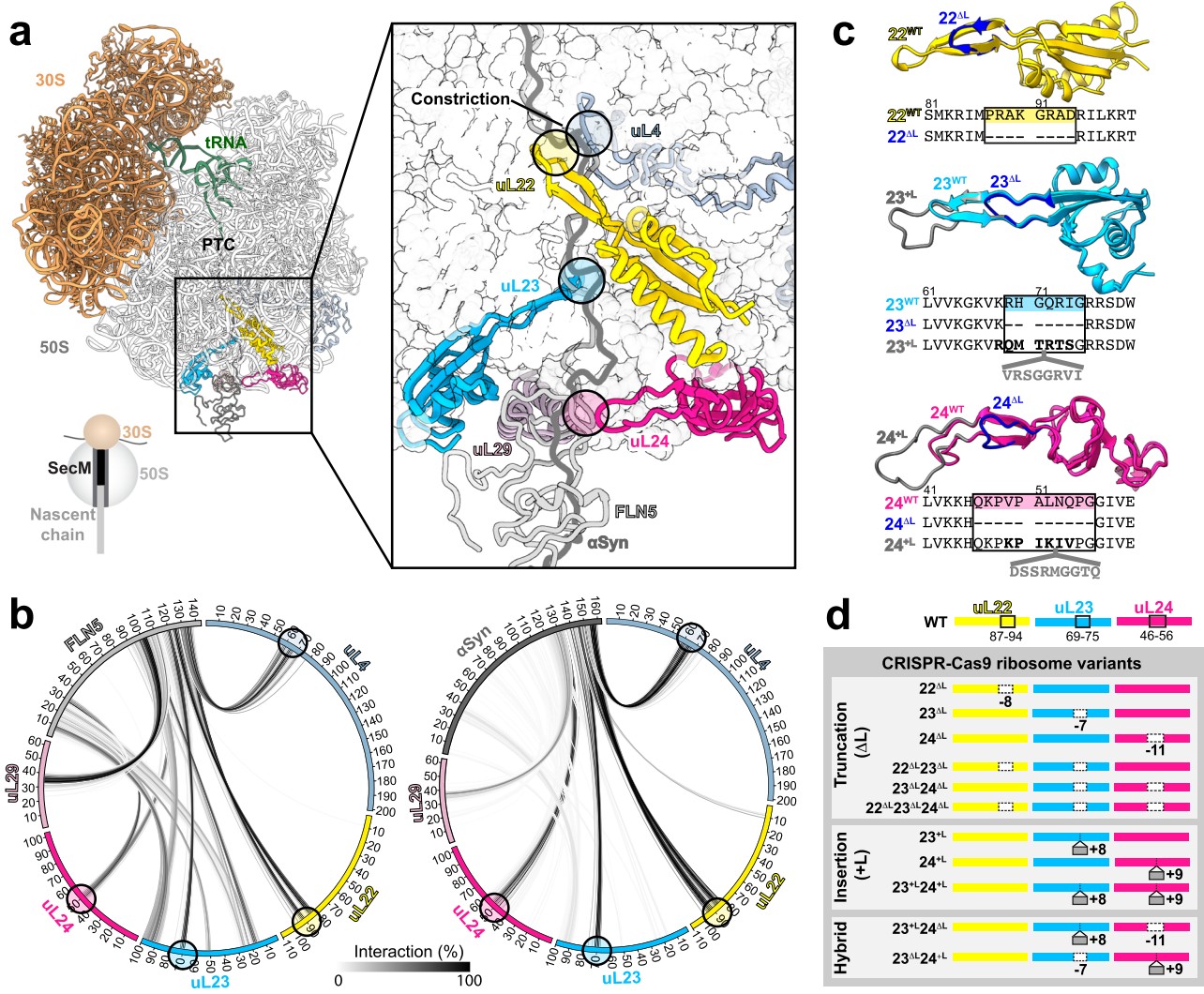

**Fig. 1 Ribosomal proteins at the exit tunnel and their structural modification by CRISPR-Cas9 gene editing. a** Structure of the 70S *E. coli* ribosome highlighting the exit tunnel (boxed). (inset) A magnified view of the loops from ribosomal proteins uL4, uL22, uL23 and uL24 that protrude into the tunnel and make contacts with FLN5 (grey) and α-synuclein (dark grey) NCs. **b** Circos plots[87], showing the interactions formed between the NCs, FLN5 (left) and α-synuclein (right), and ribosomal proteins in the exit tunnel as observed in all-atom molecular dynamics simulations restrained with chemical shifts and residual dipolar couplings[8] from NMR spectroscopy. The greyscale lines connecting NCs and the ribosomal proteins represent the extent (% of total frames) of inter-molecular interactions (see 'Methods'). The loops of uL4, uL22, uL23 and uL24 that interact with both FLN5 and α-synuclein NCs are circled. **c** Structures of uL22, uL23 and uL24 proteins from *E. coli* (PDB:3JBU) superimposed with models of truncated (ΔL, blue) and inserted (+L, grey) tunnel loops, and sequences below. **d** CRISPR-Cas9 ribosomal mutants that were generated and used in this study. Black rectangles show the loop regions that were modified.

and uL24 with loop extensions were also designed, with eight and nine residues inserted into the uL23 ($23^{+L}$) and uL24 ($24^{+L}$), respectively. These designs were drawn from two representative sequences of bacteria with exceptionally long loops (*Candidatus marinimicrobia* bacterium for uL23 and *Actinobacteria* bacterium for uL24, Supplementary Fig. 2a). In particular, the inserted loop sequence of $23^{+L}$, which comprises of positively charged residues, enabled us to explore the role of the electrostatic interactions formed between the uL23 loop and nearby rRNA segments by cryo-EM. In addition, to recapitulate the distinct vestibule environment of the eukaryotic ribosome, in which the space created by a shorter uL23 loop is compensated for by the presence of an additional protein, eL39 (Supplementary Fig. 2b), a chimeric mutant ($23^{+L}24^{ΔL}$) was created. This mutant contains extended uL23 and truncated uL24 loops and was used to delineate the possible effects arising from combining two opposing modifications on CTF. We complemented this mutant

with another opposing loop modification pair ($23^{ΔL}24^{+L}$) to further understand the interplay of the modified loops.

Using CRISPR-Cas9 gene editing via homology-directed repair[30] we generated each of the variants in BL21(DE3) *E. coli*, resulting in 11 distinct strains. We found that the editing efficiency varied widely from 1.4 to 69.2% across mutants, but its basis had no clear origin or trend (Fig. 1d, Supplementary Fig. 2d, Supplementary Table 2). We initially examined the growth characteristics of the mutant *E. coli* strains, and found that collectively they followed a trend similar to that observed for wild-type BL21(DE3) (Supplementary Fig. 2e). The exceptions were strains with ribosomes harbouring a uL22 truncation ($22^{ΔL}$, $22^{ΔL}23^{ΔL}$ and $22^{ΔL}23^{ΔL}24^{ΔL}$): these exhibited slower growth kinetics, likely as a result of this mutation imposing a delay in 50S subunit assembly[20,31]. Along similar lines, our unsuccessful attempts at truncating the uL4 loop further suggest that modifications of the constriction site have a deleterious impact

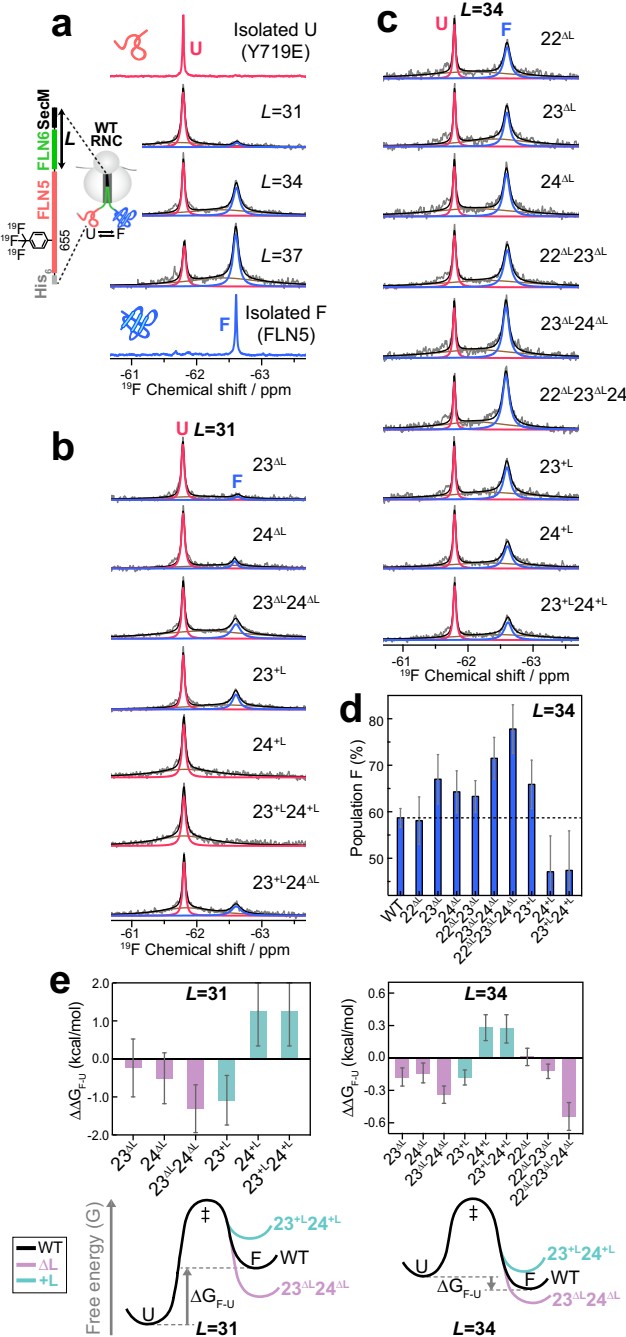

**Fig. 2 Modulation of nascent chain folding in ribosomal variants. a** 1D $^{19}$F NMR spectra of unfolded (Y719E) and folded (WT) isolated FLN5, and ribosome-bound FLN5 (WT ribosomes). For RNCs, grey spectra show the recorded spectra, while red, blue and brown peaks represent the fitted U and F state and baseline, respectively. Black spectra show the sum of the three fitted peaks. **b, c** $^{19}$F spectra of FLN5 RNCs with mutant ribosomes at $L = 31$ (**b**) and $L = 34$ (**c**). **d** Population of F state of RNCs at $L = 34$ calculated from $^{19}$F NMR analysis. The dotted line represents the F state population of the WT RNC (58%). Errors are standard deviations (s.d.) calculated by Hamiltonian Monte Carlo analysis during the fitting of real and imaginary components of the FID simultaneously. **e** The changes in the free energy of folding ($\Delta\Delta G_{F-U}$) arising from each loop modification at $L = 31$ and 34. Error is calculated by propagating the errors of $P_F$ and $P_U$ (see 'Methods'). For the variants for which no F state signal was detected ($24^{+L}$, $23^{+L}24^{+L}$, $L = 31$), maximum $\Delta\Delta G_{F-U}$ was estimated based on the assumption that $P_F < 1\%$ (See 'Methods'). Free energy landscapes of folding at $L = 31$ and 34 are shown under the bar graphs.

FLN5, either when disordered (via a Y719E mutation[6]) or natively folded, each showed a single resonance separated by a chemical shift difference of 0.8 ppm (Fig. 2a), enabling a unique method for the unequivocal analysis of both unfolded and folded FLN5 states on and off the ribosome.

FLN5 with linker lengths ($L$) of 31, 34 and 37 (Supplementary Fig. 3a) were selected as representative elongation snapshots, as these RNCs showed $^{19}$F peaks corresponding to both unfolded (U) and folded (F) states of FLN5 NCs and therefore are the most sensitive points ($L$) for detecting shifts in CTF. (Fig. 2a). These U and F states on the ribosome were in slow chemical exchange ($k_{ex} \ll |\delta_U - \delta_F| = 380\,s^{-1}$), as observed previously for isolated FLN5[33]. We also detected the presence of a highly broadened signal (Fig. 2a–c) which is consistent with intermediate states of FLN5 that have been recently observed on the ribosome[34], however, we focussed on the ground states (U and F) to accurately compare the changes in the native folding transition across different ribosomal mutants. Peak integration of the U and F states shows the population of natively folded FLN5 ($P_F$) increasing with progressive emergence, with $P_F$ of $8 \pm 7\%$, $59 \pm 2\%$ and $70 \pm 6\%$ observed at $L = 31$, 34 and 37, respectively. These observations thus show the progressive folding of FLN5 during translation.

Next, we investigated the extent of folding of FLN5 at $L = 31$ on the mutant ribosomes possessing uL23 and uL24 loop modifications. At $L = 31$, most of the FLN5 sequence has exited the tunnel during translation[6] and it is thus the earliest point in translation when FLN5 can complete its native fold (Fig. 2b). The two single truncation mutants ($23^{\Delta L}$ and $24^{\Delta L}$) showed an increased F state population ($P_F$, $11 \pm 8\%$ and $16 \pm 7\%$, respectively) relative to the WT RNC, and the double truncation increased $P_F$ even further ($42 \pm 7\%$), indicating an additive effect when both loops are shortened. By contrast, the single insertion mutants of uL23 and uL24 ($23^{+L}$ and $24^{+L}$) showed opposing effects on the folding of FLN5: enhanced folding was found in $23^{+L}$ RNCs ($P_F = 34 \pm 7\%$), whereas no F state was observed in the $24^{+L}$ RNC spectrum (Fig. 2b). The double insertion mutant ($23^{+L}24^{+L}$) also showed no F state signal, suggesting that a longer uL24 loop reduces FLN5 native folding and this cannot be rescued by extending the uL23 loop. The chimeric mutant, $23^{+L}24^{\Delta L}$, showed a significantly higher $P_F$ ($37 \pm 6\%$) relative to the WT RNCs, suggesting an additive effect of uL23 loop insertion and uL24 loop truncation which individually promote native FLN5 folding.

In $L = 34$ RNCs, which is a later point in translation, (Fig. 2c), both the single and double truncation variants of uL23 and uL24

on the ribosome and cell viability[20,31], but unknown technical limitations of CRISPR/Cas9-based gene editing may also be the source of the aberrant behaviour of this mutant.

**Co-translational folding of FLN5 on modified ribosomes monitored by $^{19}$F NMR spectroscopy.** To evaluate the impact of a redesigned exit tunnel on CTF, we monitored the extent of folding of FLN5 NCs arrested on the modified ribosomes using $^{19}$F NMR spectroscopy. The $^{19}$F isotope provides an excellent probe in large biomolecules such as RNCs due to both its high sensitivity and lack of background endogenous $^{19}$F signal in biological systems[32]. Using an orthogonal amber suppressor tRNA/aminoacyl-tRNA synthetase pair we incorporated trifluoromethyl-L-phenylalanine into position Y655 in FLN5 and recorded 1D $^{19}$F NMR spectra of both the isolated protein and the RNCs (See 'Methods', Fig. 2a–c). $^{19}$F NMR spectra of isolated

loops ($23^{\Delta L}$, $24^{\Delta L}$ and $23^{\Delta L}24^{\Delta L}$ RNCs) reveal increased $P_F$ (67 ± 5%, 64 ± 4% and 72 ± 4%, respectively) compared to WT, and follow the same trend as observed for $L = 31$ RNCs (Fig. 2c, d). Similarly, the insertion mutants ($L = 34$) also show the same opposing effects as the same variants at $L = 31$: the $23^{+L}$ RNCs show a noticeably greater $P_F$ (66 ± 5%) relative to the WT, whereas the $24^{+L}$ RNCs populate less of the F state (47 ± 8%) as does the double insertion mutant ($23^{+L}24^{+L}$, 47 ± 9%).

From the U and F populations, the free energies of folding ($\Delta G_{F-U}$) were calculated for WT and mutant RNCs at $L = 31$ and 34, and these were used to assess how the modified uL23 and uL24 loops modulate the CTF energy landscape at two different points of translation (Supplementary Fig. 3h). The changes in the free energy of folding ($\Delta\Delta G_{F-U} = \Delta G_{F-U,mutant} - \Delta G_{F-U,WT}$) show consistent results at the two linker lengths: $23^{\Delta L}$, $24^{\Delta L}$, $23^{\Delta L}24^{\Delta L}$ and $23^{+L}$ RNCs all reduce the free energy of folding ($\Delta\Delta G_{F-U} < 0$) with $23^{\Delta L}24^{\Delta L}$ showing the most negative $\Delta\Delta G_{F-U}$, while $24^{+L}$ and $23^{+L}24^{+L}$ RNCs showed positive $\Delta\Delta G_{F-U}$ (Fig. 2e). The $\Delta\Delta G_{F-U}$ values of each RNC at $L = 31$ were greater than at $L = 34$ (66 ± 23% decreased $\Delta\Delta G_{F-U}$ relative to $L = 31$), indicating that folding is modulated to a greater extent at an earlier point of translation when the NC is in closer proximity to the modified loops on the ribosome.

Unlike uL23 and uL24, the truncation of the uL22 loop shows a less significant effect on the folding of FLN5. The $^{19}F$ spectrum of $22^{\Delta L}$ RNCs is very similar to that of the WT RNCs with an identical F state population ($P_F = 58 \pm 5\%$) showing no impact on NC folding (Fig. 2c–e). The double uL22 and uL23 loop truncation variant ($22^{\Delta L}23^{\Delta L}$ ribosome) also showed no further extent of F state relative to the single uL23 truncation. The triple truncation mutant ($22^{\Delta L}23^{\Delta L}24^{\Delta L}$) showed the F state population (78 ± 5%, Fig. 2e) similar to that of $23^{\Delta L}24^{\Delta L}$ RNCs, but noticeably greater than is observed for $22^{\Delta L}23^{\Delta L}$ ribosome, again suggesting that the uL24 loop truncation has a stronger impact than that of the uL22 loop.

Overall, these NMR data reveal that modifications in uL23 and uL24 loops modulate the folding extent of the NC, whereas uL22 truncation shows a negligible influence. These data therefore suggest that the vestibule (uL23, uL24) rather than the upper region (uL22) has the most profound impact on folding.

**uL23 and uL24 loop truncation increases the spatial freedom of FLN5 nascent chains leading to an earlier folding onset.** As we found uL23 and uL24 to be major contributors to NC folding, we investigated FLN5's CTF on the $23^{\Delta L}24^{\Delta L}$ ribosomes further by generating discrete biosynthetic snapshots of increasing linker length ($L = 21-67$) that depict FLN5's U-to-F folding transition[6]. We applied a dual NMR labelling strategy to these RNCs to study the NC's uniformly-$^{15}N$ labelled peptide backbone and its selectively-$^{13}C$ methyl-labelled isoleucine side-chains (with perdeuteration)[6]. This approach allowed us to study both the U and F states of NCs, respectively, at a residue-specific level.

For FLN5 NCs on both WT and modified ribosomes, the $^1H$-$^{15}N$ HMQC spectra from $L = 21$ to $L = 37$ overlay well (Supplementary Fig. 4b) and provide ca. 80 probes of FLN5's unfolded state. Notably, there was a significantly greater reduction of unfolded peak signal intensity with increasing length in $23^{\Delta L}24^{\Delta L}$ RNCs (Supplementary Fig. 4b) compared to those of WT. This finding is mirrored in $^1H$-$^{13}C$ correlation spectra (Fig. 3a, Supplementary Fig. 5d), where RNCs from $L = 37-47$ show a markedly stronger signal intensity of folded Ile peaks in the $23^{\Delta L}24^{\Delta L}$ RNC spectra relative to WT, corresponding to the observation of greater populations of natively folded FLN5. By $L = 67$ the intensities of the F state on both ribosomes become similar, where FLN5 is at a significant distance from the ribosome. The NMR intensity profiles

for the U and F states from the $^{15}N$ and $^{13}C$ spectra (Fig. 3b) confirm an earlier folding onset for FLN5 occurring at $L < 30$ aa on $23^{\Delta L}24^{\Delta L}$ ribosomes relative to WT ($L > 37$ aa). This finding was similarly recapitulated in arrest-peptide force assays[35] (Supplementary Fig. 7) and also in all-atom MD simulations (Supplementary Fig. 8, discussed in Fig. 3d, e). In addition, the single uL23 ($23^{\Delta L}$) and uL24 ($24^{\Delta L}$) deletion mutants (inset in Fig. 3b), each showed earlier FLN5 folding ($L = {\sim}30$ aa and ${\sim}35$ aa for $23^{\Delta L}$ and $24^{\Delta L}$ RNCs, respectively) relative to WT RNC but later than $23^{\Delta L}24^{\Delta L}$ RNC, and these results complement both $^{19}F$ NMR spectroscopy (Fig. 2d) and all-atom MD simulations (Supplementary Fig. 8).

To further explore a molecular basis for the observed earlier folding mediated by the shortened uL23 and uL24 loops, we used $^{15}N$-labelled RNC ($L = 37$) samples to derive structures of $23^{\Delta L}24^{\Delta L}$ 70S ribosomes using single-particle cryo-EM. In silico sorting of the 680,941 particles yielded two major 3D classes comprising of empty 70S ribosomes, and programmed ribosomes harbouring a P-site tRNA-bound FLN5 NC (Supplementary Fig. 6). The $23^{\Delta L}24^{\Delta L}$ RNCs were resolved to an average resolution of 2.75 Å (Supplementary Fig. 6c), where the truncated uL23 loop is observed at high-resolution (local resolution 3–4 Å at 3σ, Fig. 3c) relative to the uL24 (local resolution >4 Å, at 1.8σ), which is decidedly more dynamic[36,37]. There were no other significant structural differences in these ribosomes relative to the WT 70S ribosome, suggesting that the loop modifications only altered the local environment surrounding the site of mutation. The structures of truncated loops in the empty 70S ribosome were identical to those observed in the FLN5 RNC (Fig. 3c, Supplementary Fig. 6c). Following loop truncation, there is a clear increase in tunnel volume (calculated with POVME[38]) by 1272, 1736 and 3008 Å³ for $23^{\Delta L}$, $24^{\Delta L}$ and $23^{\Delta L}24^{\Delta L}$ ribosomes, respectively. This suggests that the additional free space generated by the truncated uL23 and uL24 loops correlates with the extent of enhanced folding of FLN5 on these ribosomes as shown by NMR (Figs. 2 and 3a, b).

We complemented the cryo-EM structures with all-atom, structure-based model MD simulations by generating NC ensembles of FLN5-RNCs at $L = 31$ and 37 (Fig. 3d, Supplementary Fig. 8, Supplementary Movies 1 and 2). These analyses predict the earlier folding observed on the mutant ribosomes (Supplementary Fig. 8) and reveal the impact that the increased space in $23^{\Delta L}24^{\Delta L}$ ribosome has on the FLN5 NC (Fig. 3d, Supplementary Fig. 8). This is shown by the spatial distribution of the NC structural ensembles of WT and $23^{\Delta L}24^{\Delta L}$ RNCs which show no differences in the upper tunnel (Supplementary Fig. 8e), whereas a marked difference is observed in the lower tunnel of $23^{\Delta L}24^{\Delta L}$ ribosome. Here, the absence of the uL23 and uL24 loops generates additional space within the tunnel, thereby facilitating a shift of the most probable NC pathway towards the centre of the tunnel relative to WT RNCs, in which the NC's trajectory is diverted around the two loops (Fig. 3d). This difference in the NC structural ensemble distribution within the tunnel is further propagated outside the exit: on the WT ribosome, the loops force the folded FLN5 ensemble to nestle in close proximity to uL23, uL29 and rRNA helices (H24, H47, H50 and H59), a feature likely driven by the presence of the uL24 loop which occludes the opposite face (Fig. 3d, e). By contrast, the NC ensemble distribution is more symmetric in relation to the centre of the tunnel on $23^{\Delta L}24^{\Delta L}$ ribosomes where more contacts are made with H7 and H24 relative to WT RNCs (Fig. 3d, e). This combined cryo-EM and MD simulation analysis shows how the truncated loops provide additional space and freedom to the NC both inside and outside the tunnel.

**uL23 and uL24 loop truncation weakens the interaction of the unfolded nascent chain with the ribosome.** Recently we showed that the NC-ribosome surface interactions delay the NC folding

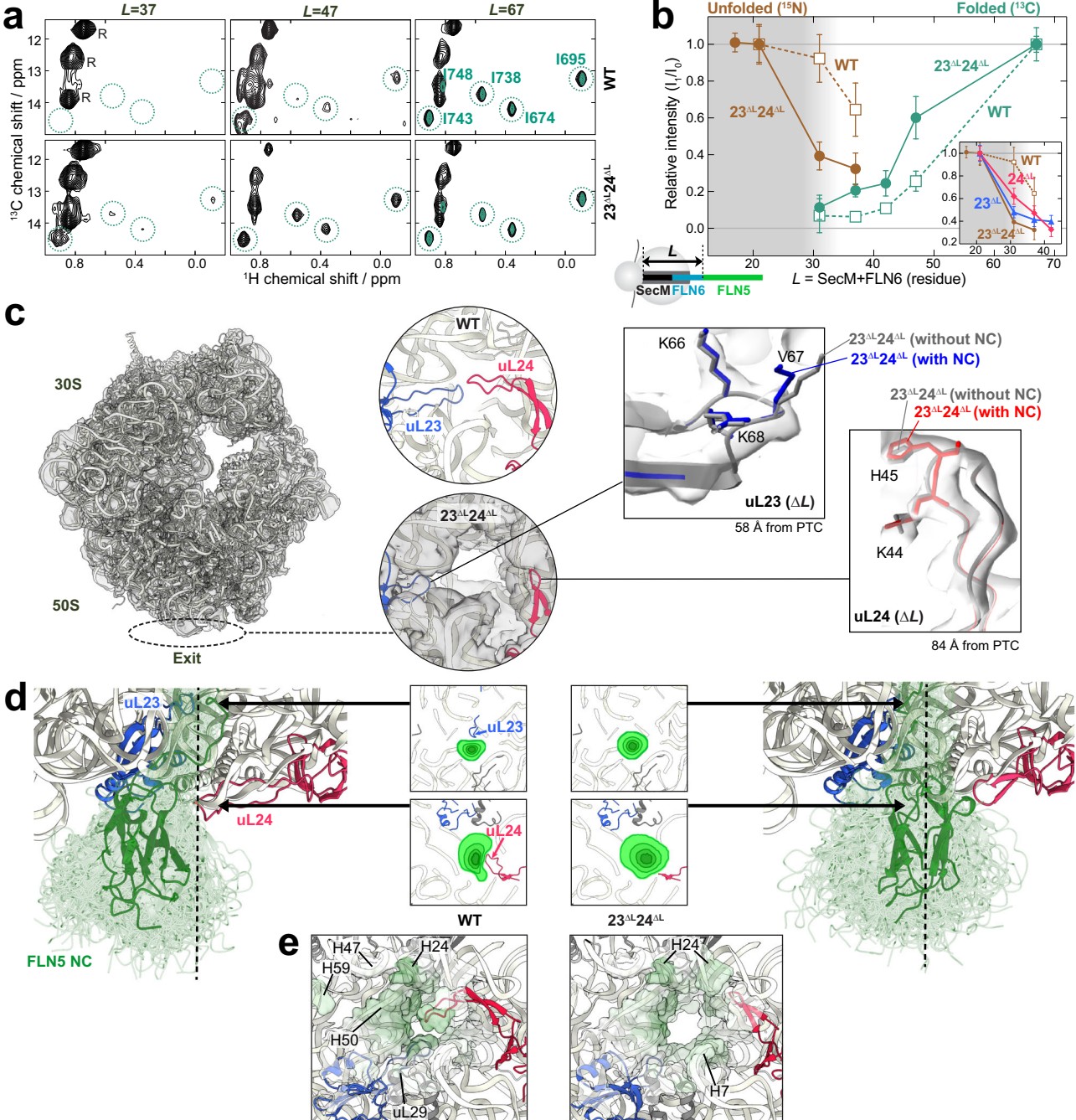

**Fig. 3 The impact of the uL23 and uL24 loop truncation on the co-translational folding of FLN5. a** $^1$H-$^{13}$C correlation spectra of folded FLN5 on WT and $23^{\Delta L}24^{\Delta L}$ ribosomes ($L = 37$–$67$). RNC spectra at $L = 67$ are overlaid with isolated FLN5 spectrum (turquoise, Supplementary Fig. 10). The Ile peaks corresponding to folded FLN5 used for intensity measurements in (**b**) (I674, I695, I738 and I743) are indicated by turquoise circles. **b** Changes in cross-peak relative intensities for three unfolded state peaks (from $^{15}$N data, corresponding to V682, A683, A694, from Supplementary Fig. 7, brown) and four folded state peaks ($^{13}$C, from **a** turquoise) within FLN5 RNCs at different $L$. Shown are WT (open square, dotted line) and $23^{\Delta L}24^{\Delta L}$ (closed circle, solid line) RNCs, respectively. Grey shading indicates the limits of the ribosome exit tunnel determined by PEG accessibility experiments[6]. (inset) relative intensities of $23^{\Delta L}$ (blue triangle) and $24^{\Delta L}$ (red diamond) RNCs. Errors are s.d. calculated by propagating the s.d. derived from the spectral noise of $^{13}$C or $^{15}$N spectra, respectively, and s.d. of nascent-chain concentrations from western blot replicates ($n = 3$). **c** Cryo-EM map of the $23^{\Delta L}24^{\Delta L}$ 70S ribosome. (circles) A magnified view of the exit tunnel from the bottom of the WT (PDB:3JBU) and $23^{\Delta L}24^{\Delta L}$ ribosome (empty ribosome cryo-EM density with the fitted uL23 and uL24 structures). (rectangles) Local density of the truncated uL23 and uL24 loops are shown with the fitted amino acids in the empty (grey) and translating (red and blue) ribosomes. **d** A structural ensemble of WT (left) and $23^{\Delta L}24^{\Delta L}$ (right) RNCs at $L = 37$ derived from all-atom MD simulations, depicting the FLN5 NC as it is present both within and outside of the tunnel (Supplementary Fig. 8). The centre of the tunnel at the exit vestibule is shown with a black dashed line. (inset) Cross-sectional views of the tunnel showing the distribution arising from the ensemble of the folded state FLN5 ($L = 37$) in the vicinity of the uL23 and uL24 tunnel loops (**e**) Ribosomal surface interaction with NC obtained from all-atom MD simulations. Interacting regions are coloured in green.

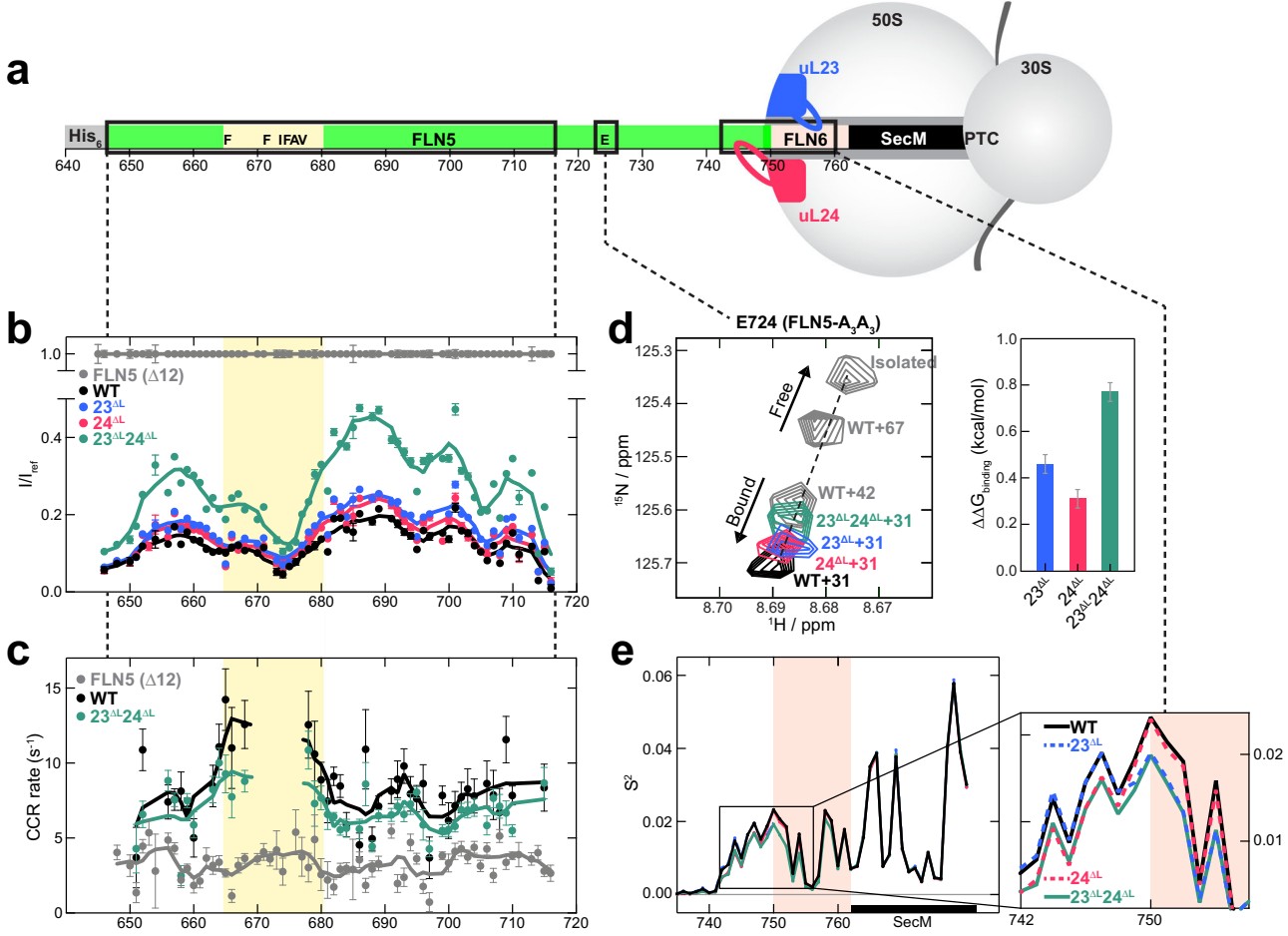

**Fig. 4 Modulation of the interaction between unfolded nascent chains and the ribosome by uL23 and uL24 loop truncation. a** Schematic of the WT + 31 RNC ($L = 31$) with sequence and domain landmarks indicated. Three regions of the NC whose dynamics was monitored by NMR and all-atom MD simulations (residues 648–717, 724 and 735–758) are shown in black rectangles. **b** Relative intensities (I) of $^1$H-$^{15}$N cross-peaks from the spectra of an isolated unfolded FLN5 Δ12 (grey circles, $I_{ref}$), WT + 21 (black circles), 23$^{\Delta L}$ + 21(blue circles), 24$^{\Delta L}$ + 21(red circles) and 23$^{\Delta L}$24$^{\Delta L}$ + 21 (turquoise circles) RNCs. A four-point moving average is plotted as a guide. Yellow-shaded region indicates the residues that interact with ribosomal outer surface. Errors are derived from spectral noise from a single measurement. **c** Cross-correlated relaxation rates of FLN5Δ12, WT + 21 and 23$^{\Delta L}$24$^{\Delta L}$ + 21. A four-point moving average is plotted as a guide. Errors are derived from the spectral noise for FLN5 Δ12 and WT, while the mean and standard error from two biological repeats are shown for 23$^{\Delta L}$24$^{\Delta L}$. **d** $^1$H-$^{15}$N cross-peaks of E724 of the A$_3$A$_3$ variant of FLN5 of the WT ($L = 31$, 42 and 67) and 23$^{\Delta L}$, 24$^{\Delta L}$, 23$^{\Delta L}$24$^{\Delta L}$ ($L = 31$) RNCs with that of the isolated unfolded protein. All spectra are recorded at a $^1$H frequency of 950 MHz. The changes in the free energy of binding ($\Delta\Delta G_{binding}$) in 23$^{\Delta L}$, 24$^{\Delta L}$ and 23$^{\Delta L}$24$^{\Delta L}$ RNCs are calculated using the populations of unbound state NC (P_Ub, Supplementary Fig. 10b). Errors are s.d. and calculated by propagating the s.d. of P_Ub. **e** S$^2$ order parameters determined from simulated ensembles of the unfolded NCs on WT, 23$^{\Delta L}$, 24$^{\Delta L}$ and 23$^{\Delta L}$24$^{\Delta L}$ RNCs ($L = 31$). The salmon pink-shaded region indicates the FLN6 linker residues between FLN5 and SecM.

by lowering its effective concentration[4]. Based on this we hypothesised that the increased spatial freedom afforded to the NC by the shortened uL23 and uL24 loops could elicit earlier folding as a result of reduced NC-ribosome interactions. To test this hypothesis we investigated the NC-ribosome interactions on the mutant ribosomes by monitoring the dynamics of different segments of the unfolded FLN5 NC (Fig. 4a).

For both WT and 23$^{\Delta L}$24$^{\Delta L}$ RNCs at $L = 21$, a similar relative intensity pattern was observed in $^1$H-$^{15}$N correlation spectra (Fig. 4b), suggesting that the nature of the local interactions of the unfolded NCs (residues 645–715) with the ribosome is broadly similar[6]. However, the 23$^{\Delta L}$24$^{\Delta L}$ RNC shows substantially stronger signal intensities relative to the WT (~120% increase), with the intensities of the single truncation variants showing modest increases (~30% and 25% for 23$^{\Delta L}$ and 24$^{\Delta L}$, respectively), indicating that the unfolded NCs are more dynamic when the uL23 and uL24 loops are truncated (Fig. 4b). $^{15}$N cross-correlated relaxation (CCR) rates of the NCs, which provide a sensitive and

quantitative measurement of NC-ribosome interactions (Fig. 4c, Supplementary Fig. 9)[4], also show a decrease for the 23$^{\Delta L}$24$^{\Delta L}$ RNC ($6.8 \pm 0.8\,\text{s}^{-1}$) relative to the WT RNC ($8.3 \pm 1.4\,\text{s}^{-1}$), indicating that the unfolded NC is on average more dynamic on the 23$^{\Delta L}$24$^{\Delta L}$ ribosomes.

We then examined the impact of the loop truncations on the C-terminal residues of FLN5 (residues 720–740 at $L = 31$), as this NC segment strongly interacts with the surface of the ribosome and regulates the onset of CTF[4]. RNCs of a folding-incompetent variant of FLN5 (A$_3$A$_3$) enabled us to quantitate the NC binding by monitoring the position of E724 resonance[4]. In contrast to the WT RNC ($L = 31$), for which the resonance of E724 is significantly downfield-shifted relative to the isolated protein due to the strong interaction of the C-terminal FLN5 with the ribosome, the 23$^{\Delta L}$, 24$^{\Delta L}$ and 23$^{\Delta L}$24$^{\Delta L}$ RNCs ($L = 31$) all show a smaller chemical shift perturbation of the E724 resonance, indicating that the interacting segment of their NC (residues 720–740) is less strongly ribosome-bound than on the WT ribosome (Fig. 4d). Based on these chemical

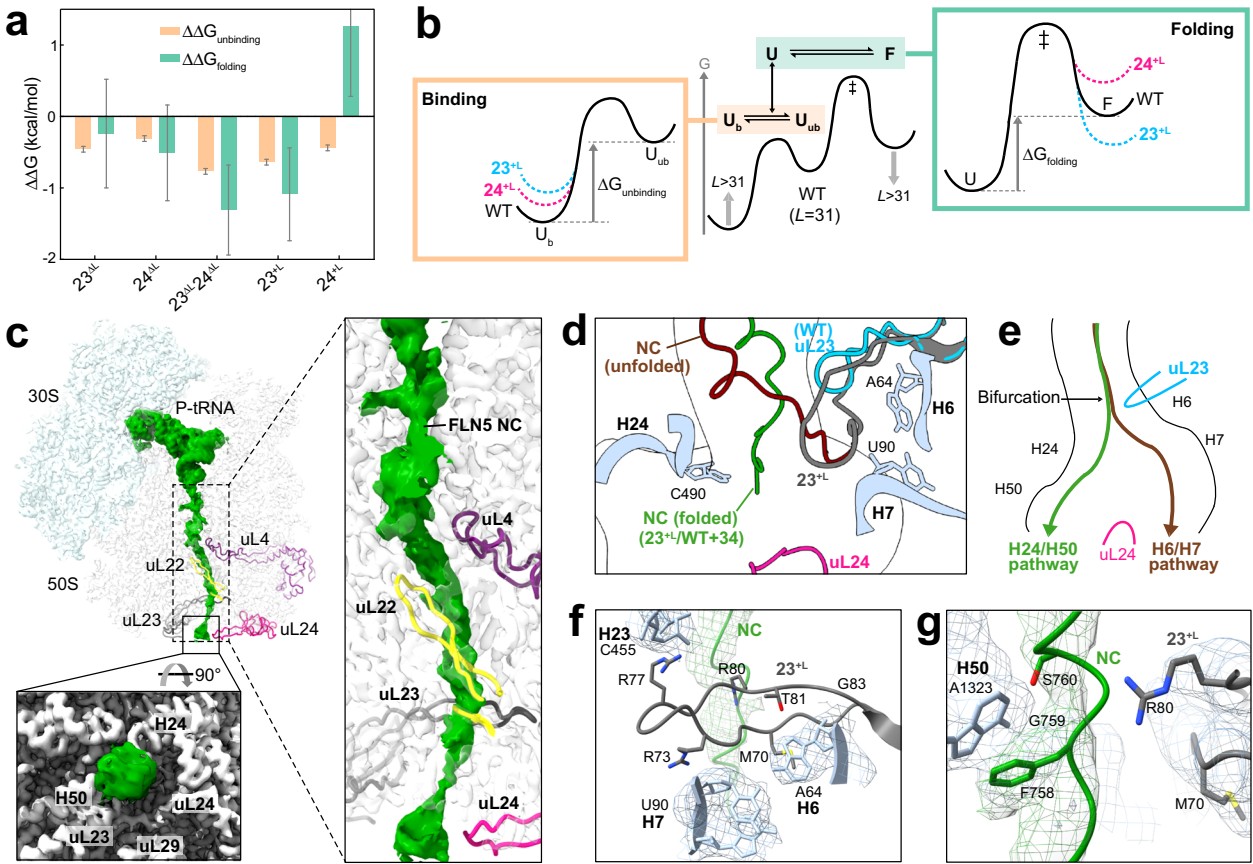

**Fig. 5 Modulation of NC binding and folding by uL23 and uL24 loop extension. a** Comparison of folding and unbinding free energies of WT and mutant RNCs ($L = 31$). The free energies and their errors are from Fig. 2e and Supplementary Fig. 10b. **b** Binding and folding free energy landscapes of FLN5 on the WT and mutant ribosomes from **a**. $U_b$ and $U_{ub}$ indicate the bound and unbound unfolded NC on the ribosome. **c–g** Cryo-EM structure of the $23^{+L}$ RNCs ($L = 34$). **c** Cryo-EM reconstruction of the $23^{+L}$ RNC. The tRNA and the NC cryo-EM density extending from the PTC through the exit tunnel is shown in green. The inset below is a view from the bottom (90° rotation) that shows the additional density corresponding to the globular domain of FLN5. The inset on the right is a close-up view of the NC density inside the exit tunnel around four ribosomal proteins. **d** Overview of the extended uL23 loop structure (grey) and its interaction with 23S rRNA and FLN5 NC (green). Unmodified uL23 loop (cyan) and FLN5 NC ($L = 47$) on the WT ribosome (brown) are shown in comparison. The schematic tunnel wall is shown in black lines and the 23S rRNA is shown in light blue. **e** Two different exit pathways of FLN5 NCs in the vestibule, bifurcated by the uL23 loop[36]. **f** Interactions between the extended uL23 loop residues and 23S rRNA. **g** Isolated cryo-EM density (green mesh) and molecular model (green cartoon) of the NC with the interacting segments from H50 and extended uL23 loop.

shift perturbations, we estimated the bound and unbound unfolded NC populations and resulting free energy of binding ($\Delta G_{binding}$) of these RNCs (Supplementary Fig. 10b)[4]. The changes observed in the difference of free energy of binding ($\Delta\Delta G_{binding}$) were $0.46 \pm 0.04$ ($23^{\Delta L}$), $0.31 \pm 0.04$ ($24^{\Delta L}$) and $0.77 \pm 0.05$ ($23^{\Delta L}24^{\Delta L}$) kcal/mol (Fig. 4d), revealing an additive effect of uL23 and uL24 loop truncations in the $23^{\Delta L}24^{\Delta L}$ RNC, a finding that was observed in the free energy of folding ($\Delta\Delta G_{folding}$) for these mutants (Fig. 2e). Furthermore, a positive correlation ($R^2 = 0.81$, Fig. 5a, Supplementary Fig. 10d) between the binding and folding free energy changes across all these RNCs was observed, suggesting that the reduced binding of the NC on these ribosome variants resulting from the truncated uL23 and uL24 loops is directly associated with the earlier folding observed on these ribosomes.

We next examined the dynamic properties of the unfolded NC within the exit tunnel: as these are NMR-invisible segments arising from both FLN5 and FLN6 (residues 741–762 at $L = 31$), we analysed the trajectories from all-atom structure-based model simulations (Fig. 3d) and estimated the rotational mobility, $S^2$ order parameters of the backbone carbonyl groups of the unfolded FLN5 NC (Fig. 4e). The FLN5 residues that are just outside of the ribosome (residues 742–750) have greater flexibility ($\Delta S^2 \sim 0.002$) in $24^{\Delta L}$ and $23^{\Delta L}24^{\Delta L}$ RNCs compared to WT and $23^{\Delta L}$ RNCs due to

the shortened loop of uL24. In contrast, residues of the FLN6 linker deeper inside the tunnel (residues 750–758, shaded in blue) show the opposite effect: greater $S^2$ values ($\Delta S^2 \sim 0.002$) were observed on $23^{\Delta L}$ and $23^{\Delta L}24^{\Delta L}$ ribosomes, suggestive of an impact of the truncated uL23 loop on this NC region.

Overall, this systematic analysis reveals that when uL23 and uL24 loops are shortened, the mobility of the disordered NC increases due to the reduced binding of two FLN5 segments (residues 665–680 and 720–750) with the ribosomal surface, and as a consequence of this lowered free energy of binding permits an earlier folding onset to develop.

**The extension of the uL23 loop changes the NC pathway within the tunnel, while the extended uL24 loop destabilises the folded NC.** Unlike the loop truncations, the extension of uL23 and uL24 loops showed opposing effects in modulating the CTF of FLN5: $23^{+L}$ caused earlier folding while $24^{+L}$ delayed folding (Figs. 2e and 5a). To investigate the molecular basis for this folding outcome, we first assessed the extent of NC-ribosome interactions by studying the binding of the C-terminal segment within the unfolded FLN5 ($A_3A_3$, residues 720–740) as in Fig. 4d. Interestingly, both $23^{+L}$ and $24^{+L}$ RNCs ($L = 31$) showed reduced binding (increased

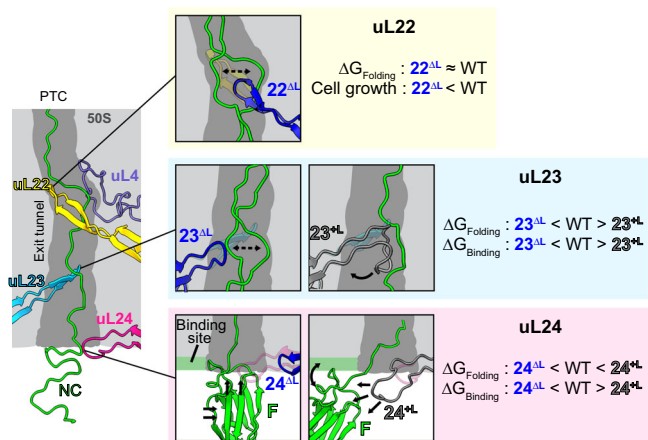

**Fig. 6 The impact of the tunnel loop modifications on the ribosome-NC interactions and co-translational folding.** Schematic illustration and the summary of the impact of loop modifications on the dynamics and folding of FLN5 NC. $\Delta G_{Folding}$ and $\Delta G_{Binding}$ denote the free energy of folding and free energy of binding of the U state, respectively.

unbinding) of the unfolded NC to the ribosome relative to the WT (Fig. 5a and Supplementary Fig. 10).

The $24^{+L}$ RNCs show increased free energy of folding despite a decrease in the free energy of unbinding, unlike all other ribosomal variants that show a good quantitative correlation between the two (Fig. 5a and Supplementary Fig. 10d). This suggests that the extended uL24 loop specifically imparts a selective and dominating destabilising effect specifically on the folded NC (Fig. 5b) which outcompetes the increased freedom of the unfolded NC. Relative to the uL23 loop, which is located further inside the tunnel, the uL24 loop occludes the wider vestibule where FLN5 folding can theoretically begin[33,39], and thus its extension reduces the conformational entropy and the structure formation of the native state FLN5, likely by reducing the space available for the folded NC. This finding is consistent with the opposing observation where the shortened uL24 loop allows earlier folding (Figs. 2 and 3b) as a result of the increased space; indeed the folded FLN5 domain is able to sample deeper inside the tunnel on the $24^{\Delta L}$ ribosomes (Supplementary Fig. 8c, Fig. 6), while no such difference was observed on the $23^{\Delta L}$ ribosome. This suggests that uL24 imparts a steric influence on the CTF mechanism.

When the folding and binding energies are compared, both the $23^{+L}$ RNCs and the truncation mutants ($23^{\Delta L}$, $24^{\Delta L}$ and $23^{\Delta L}24^{\Delta L}$) show similar trends of enhanced folding and reduced binding of the NC relative to WT RNCs (Fig. 5a). This result suggests a more complex mechanism by which the extended uL23 loop modulates the NC inside the tunnel, since its earlier folding is not elicited by increased space as in the case of the truncated uL23 loop.

To investigate how the uL23 loop extension promotes earlier folding, we analysed both $23^{+L}$ and WT RNC ($L = 34$) structures by single-particle cryo-EM. The electron density reconstruction of 547,000 and 614,463 P-site tRNA RNC particles were refined to an average resolution of 2.5 Å and 2.7 Å for $23^{+L}$ and WT RNCs, respectively (Fig. 5c, Supplementary Fig. 6). Unlike the uL23 loop in the WT ribosome that primarily interacts with 23S rRNA H6 (Fig. 5d), the extended uL23 loop interacts also with neighbouring H7 helix: the guanidino groups of R73 and R77 in the extended uL23 loop form contacts with the phosphate group of A90 (H7) and the ribose of C455, respectively, while both uL23's T81 side chain and the G83 backbone are in close proximity to A64 of H6 (Fig. 5d–f). The extended loop structure and

its interactions with H6/H7 were identical in the empty $23^{+L}$ ribosome (Supplementary Fig. 6j).

The $23^{+L}$ cryo-EM structure revealed continuous density corresponding to the P-site tRNA-bound folded FLN5 NC, which extended from the PTC throughout the entire length of the tunnel and beyond the exit (Fig. 5c, g), resembling the WT RNC structure (Supplementary Fig. 6k). Our recent cryo-EM structures of WT RNC with longer linker ($L = 45, 47$) revealed that the uL23 loop bifurcates the linker of the natively folded FLN5 NC into two discernible pathways inside the vestibule as defined by the participation of rRNA helices: the H24/H50 pathway and the H6/H7 pathway[36] (Fig. 5d, e). Notably, the folded FLN5 NC densities of $23^{+L}$ and WT RNC ($L = 34$) both show only the H24/H50 pathway, making clear interactions with two rRNA helices (A1323 of H50 and C490 of H24, Fig. 5d,g); they do not follow the H6/H7 pathway. This is seemingly a result of the short linker ($L = 34$) that can only follow the shorter H24/H50 pathway to accommodate the folded FLN5, while RNCs with a longer linker ($L = 45/47$) can take either of the two[36]. Unlike the folded FLN5 NC that follows the H24/H50 pathway, the available cryo-EM maps of different unfolded NCs with similar linker lengths ($L = 30–40$) all show their NCs primarily on the H6/H7 pathway irrespective of NC sequences[40–42], revealing the likely interactions between the unfolded NC and H6/H7. Similarly, the unfolded FLN5 NC is seen to follow the H6/H7 pathway and interact with these rRNA helices on the WT ribosome in all-atom MD simulations (Fig. 3d). On the $23^{+L}$ ribosome, however, this interaction face is occluded by the extended uL23 loop, which is highly positively charged and interacts with both H6/H7 rRNA helices, thereby reducing NC-rRNA interactions and eliciting early folding.

## Discussion

The development of an understanding of the clearly active and fundamental role of the ribosomal exit tunnel in regulating both upstream translation and downstream co-translational events for nascent polypeptides is still in its infancy[2]. We show here that by rational design and ribosome engineering we can selectively manipulate the shape of the ribosomal exit tunnel and induce modulations in CTF, which we have monitored at atomic resolution using an integrative structural and molecular biology approach. While uL22 loop truncation resulted in no noticeable change in the folding profile of FLN5 NCs, the uL23 and uL24 loop modifications showed pronounced effects on CTF upon either truncation or insertion, significantly modulating the NC folding energy landscape (Figs. 5b and 6).

Our data reveal the differential roles for the uL23 and uL24 loops in how they modulate the CTF of NCs. The uL23 loop attenuates native folding by altering the local dynamics of the NC. Moreover, by acting as a bifurcation point for the NC inside the tunnel uL23 regulates the use of the H6/H7 pathway as a further means by which it modulates NC folding beyond the exit[36]. This suggests the possibility that uL23 might be capable of relaying NC folding information from within the ribosome to its surface, an additional role to that used in SRP-dependent membrane targeting, where the uL23 loop interacts with emerging NCs to modulate SRP recruitment at the surface of the exit port[43]. The contact of the uL23 loop in *T. thermophila* mitoribosomes inside the tunnel with the SRP homolog, Ffh[44], further hints at its importance in exit port interaction energetics. By contrast, the uL24 loop alters the NC folding energy landscape by directly interacting with the folded NC and modulating its free energy. As the last structural element of the exit tunnel, the uL24 loop has a role akin to that of a gatekeeper, which is NC size-dependent: it mainly modulates the folding of proteins with complex tertiary

structures which can fold near the vestibule region, such as FLN5 (105 residues), α-spectrin (R16 domain, 109 residues) and titin (I27 domain, 89 residues)[23]. By contrast, the CTF of smaller proteins that can fold deeper inside the tunnel (ADR1a, 29 residues)[23] or alternatively, proteins that are too large to fold close to the vestibule are less likely to be modulated. In addition to the independent roles of the uL23 and uL24 loops, we have also observed that a subtle interplay exists between them in modulating the folding and dynamics of NCs: only by the truncation of both loops was there a clear additive impact on both binding and folding of the unfolded FLN5 NCs (Fig. 5a), while the extended uL24 loop alone can destabilise the folded NC (Fig. 2e).

Our structural and quantitative analysis of NC binding and folding also highlighted the influence of the rRNA on the onset of CTF. The uL23 and uL24 loop truncation variants reduced the extent of interaction between the C-terminal segment of the unfolded FLN5 and the two rRNA helices (H50, H59), and thus permit more natively folded NC to be populated. The loop extension variants similarly show reduced binding with the ribosomal surface, but different mechanisms, that modulate the folded state NC interactions with the rRNA helices inside the tunnel, are observed: the extended uL23 loop inhibits the unfolded NC interaction with H6 and H7, while the extended uL24 loop destabilises the folded NC by promoting its interaction with H24 and H50 that are present at the opposite side of the exit port. These data reveal that the NC-rRNA interactions both outside and within the exit tunnel regulate CTF, consistent with the observation that the 23S rRNA is able to interact with isolated proteins and influence their folding[45]; this characteristic of the extensive rRNA scaffold, which makes up two-thirds of the ribosome's composition, thus adds to its increasingly wider repertoire alongside translational arrest[41] and susceptibility to antibiotic targeting[46].

The exit tunnel is the sole ribosomal region that has structurally and functionally developed in all six phases of ribosomal evolution based on the accretion model[47]. The ribosomal proteins uL4, uL22, uL23 and uL24 have evolved to shape the unique geometry of the exit tunnel in different organisms, and our data reveal the relationship between evolutionary differences and CTF. This is exemplified by the chimeric mutant comprised of an extended uL23 loop and the truncated uL24 loop, a composition that closely reconstructs the exit tunnel of the eukaryotic 80S ribosome. The earlier folding observed for FLN5 on this ribosome ($23^{+L}24^{\Delta L}$) indicates an evolutionary advantage of the eukaryotic ribosome's tunnel in the manner in which it ushers NCs, which are typically larger in size and composed of >1 domain and thus have a greater requirement to fold co-translationally relative to those on the bacterial ribosome[48]. In this regard, this work opens a new avenue for exploring the structure-function relationship in ribosomes, for instance in the mitochondrial ribosome in which the uL24 loop creates another constriction site within the unusually narrow exit tunnel vestibule[49,50]; such studies also offer prospects for designing bespoke ribosomes[51]. Future investigations on the CTF of proteins with different fold types (e.g., the all alpha-helical α-spectrin) will enhance the mechanistic understanding and allow exploration of the utility of the modified ribosomes in biotechnological directions, such as enhancing the expression of recombinant proteins.

As understood from the renaturation studies of isolated proteins, protein folding is programmed by linear amino acid sequences, and protein engineering has been instrumental in dissecting these fundamental principles. Within the cellular milieu, however, these fundamental folding principles underlie an inherently multifaceted mechanism which is entirely orchestrated by the ribosome and auxiliary proteins. Merging protein engineering with the rational ribosome engineering approaches presented here, offers the capacity to provide descriptions and applications of CTF on the ribosome, as well as other folding phenomena at the point of biosynthesis, including the co-translational assembly of complexes[52] and misfolding[53]. Integrative structural biology combined with rational design thus offers an opportunity to describe such complex biological processes in an unprecedented level of molecular detail.

## Methods

**Structure-based multiple sequence alignment and design of ribosomal protein loop modifications**. Sequences of ribosomal proteins (uL4, uL22, uL23 and uL24) along with their corresponding structures from PDB (Supplementary Fig. 1) were used to build structure-based multiple sequence alignments in VMD using MultiSeq[54]. We used 8 bacterial, 2 archaeal and 19 eukaryotic sequences including 4 sequences from mitochondria and one from chloroplasts. The structure-based multiple sequence alignments were visualised in Jalview[55] and used to design the loop truncations. The designs of the uL23 and uL24 loop insertion variants were selected from the bioinformatics analysis of bacterial uL23 and uL24 sequences: all available uL23 and uL24 protein sequences, obtained from the HMMER[56] search, initiated with a single uL23 or uL24 sequence from E. coli and carried on the UniProt sequence database (https://www.uniprot.org), were clustered to 90% sequence similarity in order to remove too similar sequences and hence reduce the search space. From this list all bacterial sequences were extracted and aligned using MAFFT[57] and the sequences for the loop extension designs were selected (Fig. 1c, Supplementary Fig. 2a). Total 14,104 and 10,028 sequences of bacterial uL23 and uL24 sequences were aligned and analysed. 1257 sequences of uL23 showed insertions of four or more residues, of which two large groups of eight residue insertions with a high degree of conservation were observed. From the two groups a representative sequence, the uL23 loop sequence of Candidatus Marimicrobia bacterium (eight residue insertion, UniProt: A0A2D9QKS0) was selected and used for the extended uL23 loop sequence of $23^{+L}$ ribosome. Similar analysis of uL24 loop sequences showed 65 insertions of four or more residues, of which one group showed 5–9 residue insertions with good sequence conservation. A representative sequence of this group with the longest insertion (nine residues), the uL24 loop sequence of Actinobacteria bacterium (UniProt: A0A1F2XYG4) was selected and used for the extended uL24 loop sequence of $24^{+L}$ ribosome. Sequences with longer insertion than nine residues (11 and 13 residues) were not selected due to their poor sequence conservation.

**Ribosome modification by CRISPR-Cas9**. CRISPR-Cas9 gene editing was performed in BL21-Gold (DE3) E. coli, using a procedure as described by Jiang et al.[30] with adaptations. A two-plasmid system was used: pCas vector encoding the temperature-sensitive *Cas9* gene and $\lambda_{red}$ recombinase genes for homologous directed repair, and pTargetF vector encoding the single guide RNA (sgRNA)[30]. sgRNAs were designed using the web tool CRISPOR[58] (Supplementary Table 2). When choosing specific sgRNAs, we prioritised those that minimised both distance of cut site from the HDR junction and potential binding elsewhere in the genome (off-targets). The ribosomal genes, *rplD* (T178-A201 to be removed for the truncation design), *rplV*, *rplW* and *rplX*, were targeted by respective sgRNA to make the loop modifications of uL4 (W60-R67 for 8-residue truncation), uL22, uL23 and uL24 proteins, respectively. Donor DNA (500 ng) was provided as single-stranded DNA (Sigma or IDT) (Supplementary Table 2) complementary to the CRISPR site. Cells expressing the pCas plasmid were obtained by calcium chloride transfection and used to inoculate liquid cultures for electroporation. Liquid cultures were grown in LB and induced with arabinose at OD 0.3–0.4 to activate the expression of the λ-red recombinase system and then allowed to double in OD before preparation for electroporation. Cells were washed and incubated with donor DNA and the relevant pTargetF plasmid for 5 min before electroporation (using 0.1 cm Gene Pulser cuvettes (BioRad) at 1.8 kV, 25 mF and 200 Ohm). Cells were recovered for 1 h, plated and cultured overnight for 24–48 h. Colony PCR was used to screen for positive hits which were then identified by DNA sequencing, cured off pCas and pTargetF[30].

**Generation of ribosome–nascent chain complexes (RNCs) and isolated proteins for NMR spectroscopy and cryo-electron microscopy**. DNA constructs of SecM-stalled RNCs of FLN5 with increasing linker lengths (L) of FLN6 (L = 21, 31, 34, 37, 42, 45, 47 and 67, Supplementary Fig. 3a) are as described previously[6,59]. The WT SecM sequence[60] was used for [15]N-labelled and arrest-peptide force assay samples, whereas in cases where additional sample stability was essential ([13]C and [19]F-labelled RNCs), an enhanced translational arrest SecM sequence 'FSTPVWIWWWPRIRGPP' (AE1-SecM) was used[34]. AE1-SecM was used for all RNCs with uL22 loop truncations ($22^{\Delta L}$, $22^{\Delta L}23^{\Delta L}$ and $22^{\Delta L}23^{\Delta L}24^{\Delta L}$) to maintain efficient stalling, and we observed no release of the nascent chain («2% release after 12 h) during NMR experiments. Uniformly [15]N-labelled (U-[15]N) and [U-[2]H]-Ileδ1-[[13]CH$_3$]–labelled RNCs were generated in WT and CRISRP-modified BL21(DE3) E. coli as described previously[6]. The constructs corresponding to the unfolded FLN5 variants 'FLN5Δ12' and 'A$_3$A$_3$ FLN5' are as described in Cabrita et al.[6] and Cassaignau et al.[4], respectively. Isolated FLN5 variants were expressed and purified as previously described[61].

*[19]F-labelled samples.* For [19]F detection of RNCs, an amber suppression mutant of FLN5 (Y655Stop$_{TAG}$) was used[34]. [19]F RNCs were produced in WT and CRISPR-modified *E. coli* as described in Chan et al. Briefly, BL21(DE3) *E. coli* cells were co-transformed with the FLN5RNC (Y655Stop$_{TAG}$) vector, and also the arabinose-inducible pEVOL-pCNF-RS suppressor plasmid[62,63] which expresses the orthogonal tRNA/synthetase pair, to enable selective incorporation of the non-natural amino acid, 4-trifluoromethyl-l-phenylalanine, into the FLN5 RNC[34].

All RNCs were purified to homogeneity using a three-step protocol involving sedimentation via a sucrose cushion, followed by metal affinity and hydrophobic interaction chromatographic steps, as previously described[6,64]. All RNCs were resuspended in Tico buffer and sample concentrations were typically 5–12 μM and with OD 260 nm/280 nm ratios of >1.95. For the determination of nascent-chain occupancy, 2–5 pmol samples of RNase A–treated RNCs and an isolated FLN5 protein concentration standard were subject to low-pH electrophoresis conditions[6] and visualised by anti-histidine western blots using 1:5000 dilution of anti-6His mouse-monoclonal antibody-HRP conjugate (Invitrogen). Densitometry analysis of the RNCs were undertaken with ImageJ (http://imagej.nih.gov/ij/) software, and ensured that all sample concentrations produced signal intensities within the linear range for detection[6].

**NMR spectroscopy of RNCs and data analysis.** NMR data were acquired on an 800 MHz Bruker Avance III spectrometer (University College London) equipped with a TXI cryoprobe, and in specific cases with 800- and 950-MHz Bruker Avance III HD spectrometers (NMR Centre, Crick Institute). [15]N, [13]C and [19]F-labelled samples were recorded at 10 ([15]N) and 25 °C ([13]C and [19]F), in an interleaved manner[6]. All NMR analysis was confined to intact RNCs, as monitored using interleaved NMR diffusion/relaxation measurements and western blot[6]. [1]H-[15]N SOFAST-HMQC spectra and SORDID[65] diffusion measurements, and [1]H-[13]C HMQC spectra were recorded as previously described[6]. Methyl [1]H-T$_2$ relaxation measurements of [13]C-labelled samples were performed by recording either 1D or 2D (HMQC) spectra at two different relaxation delays (3 and 100 ms). For CCR measurements BEST-TROSY-CCR pulse scheme was used[4] at 900 (Biomolecular NMR Facility, University of Birmingham) or 950 MHz (NMR Centre, Crick Institute). For [19]F-labelled samples, multiple 1D [19]F spectra were recorded on a 500 MHz Bruker Avance III spectrometer with a recycle delay of 3 s and acquisition time of 350 ms on a Bruker Avance III spectrometer with a TCI cryoprobe[34].

All data were recorded using Topspin 3.5pl2 and initially processed and analysed with NMRPipe[66], Sparky (http://www.cgl.ucsf.edu/home/sparky/) and MATLAB (R2019b, The MathWorks Inc.). For [19]F 1D spectra, both the real and imaginary components of the FID were simultaneously fitted to Lorentzian functions with a polynomial baseline using the Bayesian information criterion. The spectra for the RNCs were fitted using two peaks for the unfolded (U) and folded (F) states as well as a broad baseline peak. For the spectra of 24$^{+L}$ and 23$^{+L}$24$^{+L}$ RNCs ($L = 31$) that showed too weak signal to be fitted into a F state peak, the maximum population of their F state (1%) was used for calculating the free energy of folding, which was estimated based on the signal-to-noise ratio of the spectrum. The free energy of folding ($\Delta G_{F-U}$) was calculated using $\Delta G_{F-U} = -RT \ln(P_F/P_U)$, and its error was calculated by propagating the errors of $P_F$ and $P_U$. The cross peaks of [15]N and [13]C spectra were assigned based on our previously deposited resonance assignments: unfolded FLN5 (BMRB:25748), folded FLN5 side-chain (BMRB:51075), FLN5A3A3 (BMRB:51023).

**Co-translational folding as monitored by [1]H-[15]N and [13]CH$_3$ correlation spectra.** CTF on the WT and modified ribosomes was measured as described in Cabrita et al.[6] with the differences as described below. The linewidths of the three well-resolved resonances (corresponding to residues A683, A694 and V682) within [1]H-[15]N correlation spectra on all the RNCs were effectively identical, but to account for the small differences peak volumes were measured and used for calculating the relative intensity of the unfolded states. Peak volumes of FLN5 + $L$ RNCs ($L = 17$, 21, 31, 37, 42) were determined by Sparky and FuDA (http://www.biochem.ucl.ac.uk/hansen/fuda), scaled for the number of scans and relative nascent chain concentrations and averaged across the three peaks[6]. For the relative intensities of [1]H-[13]C correlation spectra, four well-dispersed Ileδ1-[[13]CH$_3$]–labelled sample resonances (I674, I695, I738 and I743) of WT and 23$^{\Delta L}$24$^{\Delta L}$ RNCs ($L = 31$, 37, 42, 46 and 67) were used for analysis. [15]N intensity of WT and 23$^{\Delta L}$24$^{\Delta L}$ RNCs at $L = 21$ are used as the reference (I$_0$, fully unfolded NC) for the relative intensity of the unfolded RNCs at $L = 17$–37 (I$_1$), while [13]C intensity of $L = 67$ RNCs are used as the reference (I$_0$, fully folded NC) for the folded RNC spectra at $L = 31$–67.

**Single-particle cryo-electron microscopy of the 23$^{\Delta L}$24$^{\Delta L}$ and 23$^{+L}$ ribosomes.** Grid preparation: 250 nM of purified 23$^{\Delta L}$24$^{\Delta L}$ FLN5 ($L = 37$) RNCs and 23$^{+L}$ FLN5 ($L = 34$) RNCs were subject to cryoEM. Samples were applied (incubation time 30 and 45 s for the 23$^{\Delta L}$24$^{\Delta L}$ and 23$^{+L}$ RNCs, respectively) to glow-discharged (30 s) holey carbon grids (Quantifoil R2/1 Cu 300) pre-coated with 2 nm carbon layer. The grids were blotted for 8–9 s using Vitrobot (Thermofisher) with ambient humidity set to 100% and the temperature to 4 °C and then vitrified in liquid ethane.

Data collection and processing for 23$^{\Delta L}$24$^{\Delta L}$ FLN5 RNCs ($L = 37$): Data acquisition was performed on a Titan Krios (FEI) transmission electron microscope equipped with a Falcon III camera using EPU at 300 kV. 4328 movies were collected at a pixel size of 1.085 Å and with a defocus range of 0.7–2.2 μm. The dose was 1.1 e-/Å[2] per frame and 39 frames were aligned using Motioncor2 via Scipion[67,68] and imported to Relion 3.0[69] for further processing. CTFFIND-4.1[70] was used to determine power spectra, defocus values and astigmatism. After manual inspection the micrographs were filtered by the threshold of resolution at 5 Å resulting in 3170 micrographs. 998,921 particles (8× downscaled) were picked by Gautomatch (https://www.mrc-lmb.cam.ac.uk/kzhang/) and subjected to 2D classification. 680,941 particles were selected from the 2D classes and extracted (4× downscale) then subjected to 3D refinement using *E. coli* 70S ribosome as a reference map and low-pass filtered it to 60 Å. After 3D classification 313,668 unratcheted ribosome particles were selected and after 3D refinement, the particles were subjected to focused 3D classification with a circular mask at the intersubunit space covering P- and E-tRNA sites. Three major classes containing 102,751, 131,042 and 78,754 particles were further refined into P-tRNA RNC, empty ribosomes and P-/E-tRNA empty ribosomes, respectively. The empty ribosomal particles and the P-tRNA RNC particles were extracted without downscaling and were further subjected to postprocessing and CTF refinement for aberrations, magnification and defocus. The resulting particles were subjected to further 3D refinement and the average resolution of the final reconstruction was 2.7 and 2.75 Å for the empty and P-tRNA RNC reconstructions, respectively.

Data collection and processing for 23$^{+L}$ FLN5 RNCs ($L = 34$): Two datasets were collected using an FEI Titan Krios 300 kV electron microscope at super-resolution with a magnification of 81,000 and pixel size of 0.5335 Å using the K3 detector and EPU software. An energy slit with a width of 20 eV was used during data collection. The two datasets were combined and processed together. 15,211 movies were aligned as 7 × 5 patches with dose-weighting in RELION-3.1. After contrast transfer function estimation in CTFFIND-4.1 14,094 micrographs were selected according to the figure of merit >0.15 for further image processing. 2,075,807 particles were picked with crYOLO[71] and extracted into a 4× rescaled box size. After 2D classification, 1,826,898 particles were refined against a 60 Å low-pass filtered 70S ribosome. The first 3D classification without alignment yielded two classes of interest: empty 23$^{+L}$ ribosomes and a population of unratcheted 70 S with P-site tRNA (RNC). The latter was subjected to a focused 3D classification with an ellipsoid mask at the intersubunit space yielding 23$^{+L}$ RNCs with tRNA at the P-site. Both 23$^{+L}$ empty ribosome and RNC classes were subjected to CTF refinement and 3D refinement, and the average resolution of the final reconstruction was 2.5 Å for both empty and P-tRNA RNC reconstructions.

Ribosome and NC Modelling: For the starting molecular models of the 23$^{\Delta L}$24$^{\Delta L}$ empty ribosomes and P-tRNA RNC, we used *E. coli* 70S structure (PDB:5NWY[72]), after the tRNA and VemP structure were deleted. For the 23$^{+L}$ ribosomes and RNC we used the *E. coli* 70S structure (PDB:6YS3[40]). The initial models were rigid-body fitted into the cryo-EM reconstruction with UCSF Chimera[73]. The models were manually adjusted in COOT[74] and modified loops were manually built and real-space refined in Phenix[75] with secondary structure obtained from Phenix. Molprobity[76] was used to validate the refined model and obtain the statistics (Supplementary Fig. 6h). Local map resolution was calculated using Relion.

**Measurement of the effective volume of the tunnel.** The volume of the exit tunnel was calculated by POVME 3.0[38]. Multiple overlapped spheres with 20 Å radius are introduced inside the tunnel at the coordinates of Cα atoms of the nascent chain as reference. Grid spacing and the distance cut-off were set to 2.0 and 1.09 Å, respectively.

**Structure calculations with all-atom molecular dynamics simulations.** We used all-atom molecular dynamics (MD) simulations with the structure-based potential generated with SMOG 2.0[77]. As a model of the WT ribosome fragment used in the simulations, we chose the exit tunnel and the outer surface surrounding it from a high-resolution *E. coli* ribosome structure (PDB id: 4YBB). The surface was first chosen based on the region that directly interacts with the NC in our initial simulations (FLN5 + 45), and it was further extended by 15 Å and used for the actual simulations in order to include all potential interaction sites, in the same way as in our previous simulations of RNCs[4]. The uL23 and uL24 loop truncations were modelled in Modeller[78] and incorporated into the ribosome structure to generate the 23$^{\Delta L}$24$^{\Delta L}$ ribosome model. After we obtained the EM map of the 23$^{\Delta L}$24$^{\Delta L}$ ribosome we fitted our model into the map and found a very high cross-correlation (~0.91), which validated the quality of the model. Each NC starting structure was modelled manually inside the exit tunnel as an unfolded FLN5-FLN6 polypeptide chain and attached to the P-site tRNA via the SecM C-terminus Gly residue, which was fixed during the simulations. During simulations, the ribosome atoms were frozen, while interactions between the ribosome and NC were entirely steric in nature and modelled using the Lennard-Jones (LJ) repulsive terms. In addition, the NC native contacts used in structure-based potential were defined based on the FLN5 crystal structure (PDB: 1QFH) and were modelled using Gaussian potential[79].

For the structure-based MD simulations set up in SMOG, reduced units were applied with length, time, mass and energy scale all set to 1, except for the Boltzmann constant which is k$_B$ = 0.00831451 (kJmol$^{-1}$K$^{-1}$, default in GROMACS). As structure-based model potentials operate with reduced units,

direct comparisons of the timescale from MD with that from experiments is not possible. Since there is no direct link between the experimental temperature and that employed in MD simulations, we undertook simulations at the two different temperatures (138 and 139.5 K) found to result in the extent of folding very close to the experimental data from NMR during the FLN5 folding transition ($L = 31$–$37$). Running at 138 K also enabled isolated full-length FLN5 to remain folded, but also generated a mostly unfolded ensemble of FLN5 + 31 RNC and a mostly folded ensemble of FLN5 + 45 RNCs, consistent with the NMR observations. An enhanced sampling method Parallel Biased Metadynamics (PBMetaD[80]) was used to sample the entire free energy landscape more efficiently on the ribosome. We applied PBMetaD with multiple walkers (16), and ten collective variables (CVs) capturing the folding process: the ratio of the native contacts (Q), the radius of gyration and eight CVs describing the ratio of the native contacts but between each pair of strands: A-B, A'-G, B-E, C-F, C-C', D-E, F-F' and F-G. Gaussians corresponding to the bias potential were added every 500 steps with the height of 0.25 and the bias factor was set to 10. Simulations were run using Langevin dynamics for ca. $10^8$ time steps in GROMACS 4.5.7[81] using Plumed[82] for introducing PBMetaD. Simulations were analysed using Plumed[82], MDAnalysis[83] and VMD[84].

**Arrest-peptide force assay via in vitro translation**. Arrest-peptide force assays were performed on FLN5 RNCs as described in Ismail et al.[85], using in-house prepared WT and mutant *E. coli* S30 extracts, and cell-free translation system[19] For these experiments, DNA constructs of FLN5 RNCs[6,59], were modified to include five methionine residues after the his-tag to increase the signal from $^{35}$S methionine radiolabelling, and a 21-residue release sequence after SecM. Linear DNA fragments were amplified using T7 promoter and terminator primers[6] and in vitro transcription was carried out using T7 polymerase (NEB) with the resulting mRNA purified using RNeasy (Qiagen). Arrest-peptide force assay reactions (5.7 µL) were incubated at 37 °C with shaking at 250 rpm for 12 min and quenched with 0.5 µL of 10 mg/mL RNase. Reactions were subject to 15% Bis-Tris gels with 4x LDS loading dye and visualised using autoradiography[6]. Full length (FL) and arrested (A) band densities were quantified using ImageJ and Fityk[86], and the fraction of the full-length species ($f_{FL}$) was calculated using the equation: $f_{FL} = FL/(FL + A)$. Data reported are the average of at least two replicates.

**Monitoring growth kinetics of wild-type and mutant *E. coli* strains**. WT and ribosome-mutant BL21 *E. coli* cells were streaked onto M9-agar plates supplemented with tetracycline and grown at 30 °C for 20–24 h. 5 mL starter cultures were prepared in M9 media supplemented with tetracycline and incubated at 30 °C for 14–16 h. The cultures were then diluted 1:10 in M9 media in a volume of 1 mL and incubated for 3–4 h at 30 °C to allow the $OD_{600}$ to reach within the log phase. Cells were diluted to $OD_{600}$ of 0.01 and 100 µL was pipetted into each well (three times for each condition as technical replicates) in a 96-well plate and its growth was monitored over time at 30 °C. M9 only wells were included as blanks. Data reported are the average of at least three biological replicates performed on the same day.

**Reporting summary**. Further information on research design is available in the Nature Research Reporting Summary linked to this article.

## Data availability

All data generated in this study are available within the Article, Supplementary Data and Source Data files. Source data are provided with this paper. Five cryoEM structures have been deposited to Protein Data Bank and Electron Microscopy Data Bank with their id: $23^{+L}$: 7ZOD, EMD-14846, $23^{+L}$+34: 7Z20, EMD-14454, WT+34: 7ZP8, EMD-14850, $23^{ΔL}24^{ΔL}$: 7ZQ5, EMD-14864, $23^{ΔL}24^{ΔL}$+37: 7ZQ6, EMD-14865 Source data are provided with this paper.

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

## Acknowledgements

This work was supported by the Biomolecular NMR Facility at UCL, and also by the Francis Crick Institute through the provision of access to the MRC Biomedical NMR Centre. The Francis Crick Institute receives its core funding from Cancer Research UK (FC001029), the UK Medical Research Council (FC001029), and the Wellcome Trust (FC001029). We thank HWB-NMR staff at the University of Birmingham for providing open access to their Wellcome Trust-funded 900 MHz spectrometers. L.P.C. was supported by a UCL Overseas Research Scholarship. J.C. is supported by the Wellcome Trust (Investigator Awards 097806/Z/11/Z & 206409/Z/17/ Z). L.D.C. is funded by an AlphaOne Foundation Investigators grant and a Wellcome Trust Institutional Strategic Support Fund (UCL). The MD simulations were performed with the support of the Interdisciplinary Center for Mathematical and Computational Modeling of the University of Warsaw (ICM UW) under computational grant no. GB77-14. We acknowledge Diamond for access to the Cryo-EM facilities at the UK national electron bio-imaging centre (eBIC), proposals em20287 and bi26703, funded by the Wellcome Trust, Medical Research Council and BBSRC. We also wish to thank Dan Clare and all eBIC staff for data collection. Cryo-EM data for this investigation were also collected at ISMB EM facility (Birkbeck College, University of London) with financial support from the Wellcome Trust (202679/Z/16/Z and 206166/Z/17/Z). We thank Natasha Lukoyanova and Shu Chen for data collection and David Houldershaw for help with computing in data processing.

## Author contributions

M.A., T.W., A.M., L.D.C. and J.C. designed the project. M.A., H.S. and L.D.C. performed CRISPR/Cas9 gene editing. M.A. and H.S. performed biochemical assays. M.A., S.H.S.C., C.A.W. and A.M.E.C. acquired and analyzed the data from solution NMR spectroscopy with input from N.B., L.D.C. and J.C. T.W. performed all-atom MD simulations and analyzed the data. A.M. and E.P. acquired and analyzed the cryoEM data with input from M.A., T.W., T.A.B., R.B., L.D.C. and J.C. J.C. and L.D.C. sourced the funding and

supervised the project. M.A., T.W., A.M., L.D.C. and J.C. prepared the manuscript with input from all other authors.

## Competing interests

The authors declare no competing interests.
