## [Peer Review File · Nature Communications]

Modulating co-translational protein folding by rational design and ribosome engineeringREVIEWER COMMENTS

Reviewer #1 (Remarks to the Author):

This study uses a combination of NMR, cryo-EM and MD to investigate co-translational folding, and how protein loops in the exit tunnel affect the energetics of the process. The data reveal how protein elements of uL23 and uL24 alter co-translational folding through a combination of the steric effects. The analysis includes both, the shape of the tunnel and the interactions between the ribosome surface and the nascent chain. Simulations show how the truncated loops provide additional space for folding. Also, the aspect of the extension of the loops is of a particular interest, as uL23 and uL24 appear to have a different effect. The structures are quite high quality, and importantly, the work provides an outlook how the combination of the techniques can be used to tackle new biological problems.

Overall, this study provides several important observations that shed light on protein folding and would be of broad interest to the readership.

Minor comments:

- 1) There are some subjective descriptions and claims of novelty throughout the text, and I think the paper would benefit if those are not mentioned. For example, in the abstract “precise measurements”, “atomic resolution”, on page 3 “novel”, etc.
- 2) It would help to understand the main findings if the data, particularly MD is presented as a supplementary video. Since dynamic processes are involved, it would make the material more accessible for a reader to see it.
- 3) For structural figures, where specific residue conformations are important, such as Figure 5d-g, please consider showing the experimental density for the ribosomal elements as well.
- 4) The bifurcation of the tunnel is a really exciting feature. In a ciliate mitoribosome, a homolog of Ffh was found forming interactions with uL24m, and it might be interesting to mention in the context of the energy landscape and future directions.
- 5) Cryo-EM table usually appears as a separate table and not part of a figure, but it’s more of an editorial issue.
- 6) Figure S6g and h: would be better to show with density side by side for comparison, because it’s the experimental data that matters here.
- 7) Figure S6c: since red and blue are already used for local resolution, it might be clearer to make another panel featuring uL23 and uL24, as at the moment might look like uL23 is resolved at low resolution.
- 8) The NMR experiments are very nice.

Reviewer #2 (Remarks to the Author):

Nascent polypeptide chain (NC) folding begins as the nascent peptide is passing through the ribosome exit tunnel. While it is known that the mechanism involves interaction between the NC and parts of the ribosome, the specifics of the mechanism remain incompletely investigated. In the current manuscript, Cabrita and Christodoulou and their coworkers report a procedure for rational design of further analysis of co-translational nascent peptide folding. CRISPR was used to delete or modify loops of uL22, uL23, uL24, but the construction of uL4 mutants failed. Translation complexes were purified, and spectra were recorded for modeling nascent peptides and the structure of portions of the 23S rRNA is also reported. The results show that peptide folding involves the loops of uL23 and 24. Surprisingly, a deletion at the top of the uL22 loop does not change nascent peptide folding.

While the reported procedure is a powerful tool for the analysis of co-translational peptide folding, the story is incomplete.

1. Current evidence suggests that translational pausing and co-translational folding are coupled. Furthermore, structural changes in the PTC silence the peptide formation in response to mutations in the uL4/uL22 constriction. Genetic and biochemical evidence suggests that the uL4-uL22 constriction is important for the interaction between the nascent peptide and the ribosome exit tunnel. Hence, it is pertinent to explore: Is there an obligatory connection between pausing a nascent peptide folding? Is pausing always associated with PTC structure changes? Additional experiments should address pausing-PTC structure-folding relationships.

a. Does the current deletion in the uL22 loop affect pausing, even though it does not preclude folding? And vice versa, do the “old uL22 pausing mutants” affect folding?

b. The importance of the uL4 loop is not addressed. It is not clear why the CRISPR construction of uL4 mutants failed as spontaneous mutants are readily selected with the appropriate concentrations of erythromycin or other macrolides. The authors refer to the fact that uL4 loop contributes to ribosome biogenesis, but the loop is not essential, so their explanation is not valid. Given the known effect of uL4 loop mutations on pausing, it is essential to determine if they also affect nascent peptide folding.

2. Given the model for pausing-folding interaction, the authors should determine if the effects on folding in the uL23-24 mutants are associated with translation pausing and/or structural changes broadly in the ribosome, not just in a selected number of 23S rRNA helices. Despite the distance between the PTC and the vestibule, the tightly packed ribosome space may allow long-distance signaling.

3. Finally, a general criticism: While CRISPR is powerful for isolating specific mutants, the technique requires a preconceived notion of which parts of a molecule are most likely to be important for its function. Thus, more mutations changing different parts of the targets are necessary for a complete answer.

Reviewer #3 (Remarks to the Author):

This manuscript is a tour-de-force combining primary structure modifications through Crisper, a variety of NMR techniques to probe the nascent chain and its interactions with the ribosome, and molecular dynamics simulations to offer additional insights.

In this manuscript the authors ask how do individual proteins loops, which line the exit tunnel, impact protein co-translational folding, and binding and unbinding of the nascent chain to the ribosome? These loops have either insertions or deletions added in sizes that reflect the evolutionary variability seen across organisms. These modifications are made either individually or in combination (simultaneously present in the same ribosome).

The authors then use Fluorine NMR to probe the folded/unfolded population of protein at different nascent chain lengths on the arrested ribosome.

In this way they can identify the effects of individual loops, or synergies between them and their impact on co-translational folding. Novel findings include

1. Loops uL23 and uL24 alter co-translational folding energetics;
2. These impacts arise from changes in interactions with the exit tunnel, and the ribosome surface;
3. uL23-nascent chain interactions causes a nascent chain backbone to take one of two pathways through the ribosome exit tunnel.

This level of dissection of individual interactions and components and their impact on dynamics, binding, and folding is unprecedented, and is an exemplar of how this field of protein biogenesis biophysics can advance in the future.

That being said, I do have a few Major and minor concerns I think the authors need to address before publication.

Major concerns:

1. This paper is quantitative, and its conclusions are based on quantitative analyses. Yet, the paper falls short of acceptable standards for statistical analysis of hypotheses. And obtaining these statistics has the potential to change some of their qualitative interpretations.

Specifically,

1.1 In each figure the caption should specify what the error bar represents (Confidence interval, standard deviation, or something else.) The number of replicates, p-values and the statistical test used should be reported (or *, **, *** standard in molecular biology should be reported to give the range of p-values).

1.2 Fig. 2e (illustrative free energy profiles) - it is not a statistically justified interpretation to draw different profiles as different if you cannot reject the null hypothesis that the energy difference is not different than zero. Since your error bars overlap with zero (WT) for $23^{\Delta L}$ and $24^{\Delta L}$, it is not justified to draw the free energy profile for these two systems as not overlapping with the WT free energy profile. This needs to be corrected.

1.3 Fig 5a,b, the same mistake as above is made in the free energy profiles in this graph.

1.4 p-values (or stars) need to be put above each bar in the bar plots in Fig. 2e, Fig. 5a, Fig. 4d, Fig. 2d (where the statistical test is if the P_F is statistically different than wildtype), and anywhere else such data are presented in the main text or SI.

2. The Methods section do not state the temperature at which the simulations were, run, this information is stated in the caption of an SI figure to be around 150 K. I assume the NMR experiments were done at room temperature (290 K). This temperature discrepancy is unexplained, and seems unjustified. The simulations should probably be rerun at the experimental temperature, or a compelling, physical reason (that I can't imagine right now) must be given as to why it is appropriate to compare simulations at 150 K to experiments at 290 K.

3. Through the use of particular adjectives throughout the paper, there seems to be an overselling of the implications of this work. Here are a few examples:

3.1 line 75: "these loops form the basis of novel regulatory mechanisms that modulate the folding energy landscape of NCs."

The fact that you perturb a molecule (the ribosome), and another molecule changes its behavior (the nascent chain) does not demonstrate a biological regulatory mechanism. To go beyond a mere physical interaction (which the authors have shown) to a regulatory mechanism, there needs to be a demonstration of an evolutionarily evolved adaptation mechanism of form or behaviour of an organism or

subcellular process to changed conditions. The authors have not shown this, so the wording about regulatory mechanisms should be removed.

3.2 line 108: "shaping the trajectories of emerging NCs". Instead of shaping, a more scientific and less loaded term would be 'influencing'.

4. To back up the claim in Line 340, the P_F from the simulations (Fig. S8) versus the experimental P_F should be plotted against each other as another panel in Fig. S8 and a correlation and p-value reported. (This will depend on the temperature issue raised earlier, as well).

5. Line 529: "This suggests that uL23 can also relay NC folding information..." I think the phrasing of this claim is too strong given the evidence (Logic: there is different behavior of nascent chains interacting with uL23 variants, so therefore uL23 can relay information). I think a more appropriate claim (given the data) would something like "This suggests the possibility that uL23 might be capable of relaying NC folding..."

Minor:

line 134: Provide the loop lengths when referencing these: "to those observed in archaeal and eukaryotic (cytosolic) ribosomes"

The equivalent of Fig. 2d needs to be shown for $L=31$. SI is fine.

Line 351: What does 'centrosymmetric' mean?

Line 562: Can you explain what the "six phases of ribosomal evolution" are? Sounds interesting.

Line 48: The sentence reads strange to me by ending it with the words 'these'. Perhaps this is subjective, and the authors are free to ignore this comment if they wish.

Reviewer #4 (Remarks to the Author):

Ahn et al describe a thorough and impressive study of the co-translational protein folding process that occurs in the ribosome exit tunnel. The study applies very powerful NMR tools to monitor the folded and unfolded states of the nascent chain as well as the interactions of the chain with the exit tunnel. Using these tools, they explore the engineering of three important protein loops of the ribosome that interact with the nascent chain as it folds. They employ CRISPR/Cas9 methods to re-engineer these loops making deletions and additions, and they utilize cryoEM structures of the engineered variants combined with molecular dynamics simulations to illustrate the role that the loops within the tunnel play in folding the nascent chain. The data are quite remarkable, and the arguments are convincing.

The detailed elucidation of the L4, L22, L23, and L24 loops in assisting or manipulating folding events is quite impressive. The arguments wherein L23 and L24 work cooperatively is supported by the ability to invert the function through engineering. The combination of cryoEM structures supporting the engineered loops with the use of focused molecular dynamics, provides the detailed insight 'within' the tunnel that has otherwise been nearly impossible to discern. The correlation of the relative function of the loops and their evolution across different organisms extends fundamental understanding of protein expression and suggests opportunities for design. The authors seem to suggest that design of modified ribosomes may be beneficial in some biological context. This seems rather oblique as modification or re-engineering of an organism's ribosome (particularly a human) would seem to bias away from productive biology and be unmanageable for the full breadth of protein expression required of the organism. Nevertheless, the exquisite detail that these studies indicate is possible becomes a pathway for mechanistic understanding that may eventually link with sequence design to facilitate biological advantage. It will be intriguing to see if future engineering can bias the productive expression of certain fold-types.

Questions/More info needed:

It is not clear to what extent the particular FLN5 CTF is selected for timescales of interaction. Will other systems be amenable or is this a special case? This may fall into categories of fold-type nascent chains, and any comments on this would be welcome.

It is intriguing that the ^{19}F measurements correspond to slow exchange and the ^{15}N measurements suggest faster averaging. Can the authors be more explicit as to the timescales that the measurements are sensitive to and how this compares with the timescales suggested in the MD simulations?

There is no indication of the deposition of structural data to the Protein Data Bank. This should be done for the cryoEM structures determined.

The chemical shift assignments may have been previously reported. If so, then the BMRB accession number should be provided. If not, then a deposition to the BMRB of the assigned chemical shifts should be accompanied with this publication.

Minor points/corrections

Page 12, upper paragraph: Suspect that the authors mean 1.8 sigma, and the font is not set to symbol.

Page 14, last line: What is the length (in time) of the molecular dynamics simulations? The manuscript indicates 10^8 steps, in Langevin dynamics, which does not correlate well with real time. What is the estimate of the time for comparison?

Supplementary Information

1. Page 3, first paragraph: reference is made to Chan et al. but there is no citation information. Please update.
2. Page 3, NMR spectroscopy: At what field strength were the ^{19}F spectra acquired?
3. Would like to see an example equation for computing ΔG (and $\Delta\Delta G$) from populations in NMR spectra. This would provide better confidence and understanding to readers from different backgrounds.

REVIEWER COMMENTS:

Reviewer #1 (Remarks to the Author):

This study uses a combination of NMR, cryo-EM and MD to investigate co-translational folding, and how protein loops in the exit tunnel affect the energetics of the process. The data reveal how protein elements of uL23 and uL24 alter co-translational folding through a combination of the steric effects. The analysis includes both, the shape of the tunnel and the interactions between the ribosome surface and the nascent chain. Simulations show how the truncated loops provide additional space for folding. Also, the aspect of the extension of the loops is of a particular interest, as uL23 and uL24 appear to have a different effect. The structures are quite high quality, and importantly, the work provides an outlook how the combination of the techniques can be used to tackle new biological problems.

Overall, this study provides several important observations that shed light on protein folding and would be of broad interest to the readership.

We thank the reviewer's appreciative comments.

Minor comments:

1) There are some subjective descriptions and claims of novelty throughout the text, and I think the paper would benefit if those are not mentioned. For example, in the abstract "precise measurements", "atomic resolution", on page 3 "novel", etc.

We have removed several subjective descriptions including those on Page 1 ('precise', 'at atomic resolution', 'important' and 'useful' are removed), Page 3 ('form the basis of novel regulatory mechanism that' is removed) and Page 20: ('unique' is removed)

2) It would help to understand the main findings if the data, particularly MD is presented as a supplementary video. Since dynamic processes are involved, it would make the material more accessible for a reader to see it.

We welcome this suggestion and have generated movies (Movie S1, Page 9) depicting the structural ensemble from the all-atom MD simulations of FLN5 on both WT and 23^{ΔL}-24^{ΔL} ribosomes.

3) For structural figures, where specific residue conformations are important, such as Figure 5d-g, please consider showing the experimental density for the ribosomal elements as well.

We have revised Figures 5f and 5g (page 16) to also depict the electron density of the ribosomal elements – both protein (uL23 loop) and rRNA helices (H6, H7, H24 and H50). The density of the extended uL23 loop is additionally shown in Figure S6i.

4) The bifurcation of the tunnel is a really exciting feature. In a ciliate mitoribosome, a homolog of Ffh was found forming interactions with uL24m, and it might be interesting to mention in the context of the energy landscape and future directions.

We appreciate the reviewer's suggestion. The following sentence and reference has been added to the Discussion (p.18):

The contact of the uL23 loop in *T. thermophila* mitoribosomes inside the tunnel with the SRP homolog, Ffh⁴⁵, further hints at its importance in exit port interaction energetics.

⁴⁵Tobiasson V, Amunts A. Ciliate mitoribosome illuminates evolutionary steps of mitochondrial translation. *Elife*, e59264 (2020).

5) Cryo-EM table usually appears as a separate table and not part of a figure, but it's more of an editorial issue.

We have added these data to a table (Table S1, Page 8)

6) Figure S6g and h: would be better to show with density side by side for comparison, because it's the experimental data that matters here.

We now depict the experimental density together with the modelled structure for the extended uL23 loop in Figure S6i (SI, page 18).

7) Figure S6c: since red and blue are already used for local resolution, it might be clearer to make another panel featuring uL23 and uL24, as at the moment might look like uL23 is resolved at low resolution.

We have added a new panel, Fig S6c with the truncated uL23 and uL24 loops coloured by their local resolution (SI, page 18).

8) The NMR experiments are very nice.

We thank the reviewer for the positive comment.

Reviewer #2 (Remarks to the Author).

Nascent polypeptide chain (NC) folding begins as the nascent peptide is passing through the ribosome exit tunnel. While it is known that the mechanism involves interaction between the NC and parts of the ribosome, the specifics of the mechanism remain incompletely investigated. In the current manuscript, Cabrita and Christodoulou and their coworkers report a procedure for rational design of further analysis of co-translational nascent peptide folding. CRISPR was used to delete or modify loops of uL22, uL23, uL24, but the construction of uL4 mutants failed. Translation complexes were purified, and spectra were recorded for modeling nascent peptides and the structure of portions of the 23S rRNA is also reported. The results show that peptide folding involves the loops of uL23 and 24. Surprisingly, a deletion at the top of the uL22 loop does not change nascent peptide folding. While the reported procedure is a powerful tool for the analysis of co-translational peptide folding, the story is incomplete.

We thank the reviewer for their comments that highlight some key considerations that we address more fully below.

Fundamentally the manuscript is a high-resolution structural and biophysical analysis of 11 newly engineered ribosomes. It shows over 50 RNCs analysed by NMR (ca. 10 mg of purified selectively-labelled RNCs required for each) and 6 cryoEM structures – this is a very major undertaking. The molecular and structural biology framework created allows us to describe, for the first time, a detailed residue-specific understanding of the impact of ribosomal elements on structure, dynamics and folding of nascent chains. Protein folding has, until very recently, only been described at this resolution for isolated proteins while here we describe these phenomena in thermodynamic and structural detail on the ribosome. We therefore consider this work to be a major advance showing that integrative structural biology coupled with rational design will enable us to address increasingly complex co-translational questions at a molecular-level.

While we described the impact of tunnel protruding loops of uL22, uL23 and uL24 on the CTF, we have been unsuccessful with uL4 loop modification by CRISPR/Cas9. Nevertheless, we believe that the findings that we present based on the rational loop insertion/truncations of uL23 and uL24 are highly relevant insights that have not been considered previously. The negligible impact of the uL22 loop variation on co-translational folding may reveal that this upper tunnel loop does not alter the free energy landscape of folding within the vestibule under equilibrium when the translation elongation rate or pausing is not affected. This is an interesting question that certainly deserves systematic investigation in a separate, follow-up study.

1. Current evidence suggests that translational pausing and co-translational folding are coupled. Furthermore, structural changes in the PTC silence the peptide formation in response to mutations in the uL4/uL22 constriction. Genetic and biochemical evidence suggests that the uL4-uL22 constriction is important for the interaction between the nascent peptide and the ribosome exit tunnel. Hence, it is pertinent to explore: Is there an obligatory connection between pausing a nascent peptide folding?

The Reviewer has highlighted some significant considerations, including the relationship between translational pausing and co-translational folding and the role of ribosome-nascent chain cross-talk; we have been discussing these ideas within the laboratory for some time and they are deserving of stand-alone studies which we are actively pursuing in the context of work on nascent chain misfolding of alpha-1-antitrypsin. However, the current *equilibrium* study to consider the impact of loops on NC folding harnesses SecM-mediated stalling, which establishes a stable pausing that has no influence on the NC conformations observed.

Whether pausing/stalling *facilitates* nascent peptide folding is a more complex question. To produce equivalent samples for both NMR and cryoEM, we fused the SecM arrest motif at the C-terminus of nascent peptide of increasing lengths to create “biosynthetic/folding snapshots” and produced the samples with *E. coli*; the SecM sequence itself starts at the PTC and ends just below the uL4/uL22 constriction site. The presence of SecM means that we effectively decouple a nascent peptide’s folding status from pausing/stalling effects because each RNC has been arrested in the same way. Our study therefore does not seek out whether there is an obligatory connection between pausing and nascent peptide folding. Instead, our study shows how we are able to quantify the extent of co-translational folding that can be achieved by a nascent peptide when it is bound to the ribosome. We have presented a framework which enables us to quantify the structural preferences of the

emerging nascent polypeptide within an altered ribosomal environment, and we are thus now poised to embark on the types of (excellent) questions posed by the Reviewer.

Is pausing always associated with PTC structure changes? Additional experiments should address pausing-PTC structure-folding relationships.

SecM stalling motif does not induce any noticeable structural changes to the PTC as it severely stalls the nascent polypeptide, except for a small change in the geometry of P-site peptidyl-tRNA (2Å), while even less change was observed for another stalling sequence TnaC (Bhushan *et al.*, <https://doi.org/10.1371/journal.pbio.1000581>). We analysed our cryoEM structures, which have higher resolutions than those described in the above paper, to investigate this aspect further and added new figure panels (S6I,m), which reveal 1) effectively identical PTC structures of the ribosome with and without SecM-NC (RMSD=0.4Å), and further 2) no structural changes of PTC across different RNCs (RMSD=0.21Å). This demonstrates that SecM-induced pausing does not elicit structural changes in the PTC to influence NC folding outcome. We cannot, however, specifically comment on whether pausing/stalling is generally associated with PTC changes in other existing stalling sequences and this is beyond the scope of this paper, but it remains a fascinating question.

a. Does the current deletion in the uL22 loop affect pausing, even though it does not preclude folding? And vice versa, do the "old uL22 pausing mutants" affect folding?

The Reviewer's comment is rightly raised in that the SecM-induced stalling has been shown to be compromised by the mutations (residues in G91 and A93, Nakatogawa *et al.*, [https://doi.org/10.1016/S0092-8674\(02\)00649-9](https://doi.org/10.1016/S0092-8674(02)00649-9)) and small insertions or deletions (Lawrence *et al.*, <https://doi.org/10.1128/JB.00632-08>) in the uL22 loop region. As the cell-based assays in these studies indicated, it is likely that the structural variations in the uL22 loop can alter translational pause in the cell, thereby affecting the folding of nascent polypeptides.

More recently published papers that delineate the mechanism of SecM-mediated stalling reveal that the stalling indeed involves the interaction of uL22 loop with the SecM nascent chain (Bhushan *et al.*, <https://doi.org/10.1371/journal.pbio.1000581>, Zhang *et al.*, <https://doi.org/10.7554/eLife.09684.001>). Because of these aspects, in our ¹⁹F NMR measurements of the RNC with uL22 loop deletion, we used a mutant version of SecM (AE1-SecM) that enhances the stability of the RNCs (days) relative to wild-type SecM (hours), and comparable to the other ribosomal mutants that we made (<<2% release after 12hr). Interestingly, a recent study with a longer deletion of uL22 (17 residues) than ours (8 residues) showed only negligible effect on the wild-type SecM stalling efficiency (Lawrence *et al.*, <https://doi.org/10.1093/nar/gkw493>). These data collectively indicate that our uL22 loop deletion mutant does not perturb translational pausing of the nascent chain. For the reasoning mentioned above (use of AE1-SecM), we also anticipate that the "old uL22 pausing mutants" would not affect *equilibrium* folding.

b. The importance of the uL4 loop is not addressed. It is not clear why the CRISPR construction of uL4 mutants failed as spontaneous mutants are readily selected with the appropriate concentrations of erythromycin or other macrolides. The authors refer to the fact that uL4 loop contributes to ribosome biogenesis, but the loop is not essential, so their explanation is not valid. Given the known effect of uL4 loop mutations on pausing, it is essential to determine if they also affect nascent peptide folding.

Based on the CRISPR/Cas9 gene editing protocol that we utilised, we narrowed down to three potential reasons for the lack of success with uL4 loop modification: 1) any secondary structure of genomic DNA (*rplD*) that precludes the Cas9 binding and the introduction of double-stranded break (DSB). 2) lower Homology-Directed Repair (HDR) efficiency due to the design of donor ssDNA. 3) cell-growth defect due to slow ribosome assembly. To explore 1), we tested gene editing without donor ssDNA, but could not clearly confirm the efficiency of Cas9 for making DSB. For 2) we tested several different loop designs – different Protospacer Adjacent Motif (PAM) sites, different lengths of deletion (-4 and -8), different lengths and symmetry of homology arms in the donor ssDNA. To test 3), we also harnessed the advanced CRISPR/Cas9 gene editing technique for *E. coli* without temperature sensitivity (Li *et al.*, <https://doi.org/10.1093/abbs/gmab036>) for screening slowly growing transformants. Unfortunately, none of these resulted in successful transformants, so currently it is not clear why the mutation is not achieved by CRISPR/Cas9 gene editing.

As the Reviewer pointed out, both uL4 and uL22 loops were shown to be not essential for ribosome assembly by Janice Zengel *et al.* (<https://doi.org/10.1261/rna.5400703>), but the same group's more recent study (<https://doi.org/10.1093/nar/gkw493>) also showed that the truncation of both loops *do affect* ribosome assembly. In our own studies, we observed that the uL22 loop truncation results in a cell-growth defect, which we therefore anticipate that this result likely reflects slower ribosome assembly. Based on the Reviewer's comment we have changed the text as follows:

From:

"Along similar lines, our unsuccessful attempts at truncating the uL4 loop, which also has a known role in 50S subunit assembly, further suggest that modification of the constriction site have a deleterious impact on the ribosome and cell viability"

To:

"Along similar lines, our unsuccessful attempts at truncating the uL4 loop further suggest that modifications of the constriction site have a deleterious impact on the ribosome and cell viability, *but it is not clear why CRISPR/Cas9-based gene editing failed with this mutant.*"

2. Given the model for pausing-folding interaction, the authors should determine if the effects on folding in the uL23-24 mutants are associated with translation pausing and/or structural changes broadly in the ribosome, not just in a selected number of 23S rRNA helices. Despite the distance between the PTC and the vestibule, the tightly packed ribosome space may allow long-distance signaling.

We have demonstrated by western blot and NMR diffusion/relaxation measurements that both uL23 and uL24 loop mutant ribosomes show no release of nascent chain, confirming the integrity of stable stalling as on the WT ribosome. In our cryoEM structures of modified ribosomes (23^{ΔL}24^{ΔL} and 23^{+L}) we have not observed any significant structural changes of rRNA or ribosomal proteins throughout the exit tunnel including the PTC and outer surfaces. Our newly added figure panel (Figure S6m) clearly demonstrates that there is no structural impact on the PTC by the uL23 loop modification (RMSD=0.21 Å).

3. Finally, a general criticism: While CRISPR is powerful for isolating specific mutants, the technique requires a preconceived notion of which parts of a molecule are most likely to be important for its function. Thus, more mutations changing different parts of the targets are necessary for a complete answer.

We agree with the Reviewer that there is great value in using selective evolutionary pressure, which has the unique advantage of potentially uncovering new mechanistic principles without little to no information *a priori*. This approach has enabled, for example, the identification of novel antibiotic binding sites (Wekselman *et al.*, <https://doi.org/10.1016/j.str.2017.06.004>) and insights into ribosome assembly (<https://doi.org/10.1093/nar/gkw493>). In our study, we have embarked on a set of hypothesis-driven questions which have come from a structure- and bioinformatics-led analysis of our RNC structures. CRISPR/Cas9-based editing enabled us to introduce site-specific modifications in a controlled manner (akin to site-directed mutagenesis used in protein engineering studies), and we believe that this rational design approach of the ribosome is the strongest part of our methodology (which has also not been reported elsewhere before). Our rational design approach has also enabled us to reveal new observations and our chimeric mutants of uL23 and uL24 loop insertions clearly demonstrate this: we were able to selectively introduce sequences of two bacterial species (*Candidatus marinimicrobia* bacterium for uL23 and *Actinobacteria* bacterium for uL24) into the *E. coli* 70S ribosome (23^{+L}, 24^{+L}), and our subsequent cryoEM structure of the 23^{+L} ribosomes reveals a novel uL23-H7(23S RNA) interaction.

Reviewer #3 (Remarks to the Author)

This manuscript is a tour-de-force combining primary structure modifications through Crisper, a variety of NMR techniques to probe the nascent chain and its interactions with the ribosome, and molecular dynamics simulations to offer additional insights.

In this manuscript the authors ask how do individual proteins loops, which line the exit tunnel, impact protein co-translational folding, and binding and unbinding of the nascent chain to the ribosome? These loops have either insertions or deletions added in sizes that reflect the evolutionary variability seen across organisms. These modifications are made either individually or in combination (simultaneously present in the same ribosome).

The authors then use Fluorine NMR to probe the folded/unfolded population of protein at different nascent chain lengths on the arrested ribosome.

In this way they can identify the effects of individual loops, or synergies between them and their impact on co-translational folding. Novel findings include

1. Loops uL23 and uL24 alter co-translational folding energetics;
2. These impacts arise from changes in interactions with the exit tunnel, and the ribosome surface;
3. uL23-nascent chain interactions causes a nascent chain backbone to take one of two pathways through the ribosome exit tunnel.

This level of dissection of individual interactions and components and their impact on dynamics, binding, and folding is unprecedented, and is an exemplar of how this field of protein biogenesis biophysics can advance in the future.

We thank the reviewer for their appreciative comments.

That being said, I do have a few Major and minor concerns I think the authors need to address before publication.

Major concerns:

1. This paper is quantitative, and its conclusions are based on quantitative analyses. Yet, the paper falls short of acceptable standards for statistical analysis of hypotheses. And obtaining these statistics has the potential to change some of their qualitative interpretations.

We thank the Reviewer for their comments on our statistical analyses; these form a crucial part of our activities when dealing with large complexes in solution with inherently limited lifetimes. Very significant amounts of measurement time (often at the expense of the main experiment signal) are indeed spent to ensure sample integrity and validity of our interpretations. e.g., via diffusion-NMR experiments alongside biochemical analyses over time. We have described such analyses previously (Cassaignau *et al.*, <https://doi.org/10.1038/nprot.2016.101>) and the mode of assessing our data is continuously undergoing scrutiny and improvement. Importantly however, structural biology experiments such as NMR/cryoEM/X-ray crystallography do not typically involve averaging across multiple biological or technical replicates, and thus many of the statistical tests listed by the Reviewer (in points 1.1, 1.2 and 1.4) cannot be faithfully applied in the same way as for molecular biology/cell biology/quantitative biochemistry experiments. RNC samples *are all repeatedly prepared* in the laboratory; we find high reproducibility across identically prepared samples (both technical and biological repeats).

We have performed all statistical analyses as is the standard in the field: i.e., NMR involves a statistical analysis of the signal:noise ratio within a dataset with values cited with at least 1 standard deviation. For cryoEM structures, the overall resolution of our reported structures has been derived from a Fourier Shell Correlation of 0.143 (gold standard (Rosenthal *et al.*, <https://doi.org/10.1016/j.jmb.2003.07.013>), and as needed, we also cite the sigma level (i.e. confidence level) when reporting any local electron density (Pages 12 and 17 in the revised manuscript).

Wherever possible we have now included more statistical information within the Methods and Figure legends as detailed below for points 1.1, 1.2 and 1.4.

Specifically,

1.1 In each figure the caption should specify what the error bar represents (Confidence interval, standard deviation, or something else.) The number of replicates, p-values and the statistical test used should be reported (or *, **, *** standard in molecular biology should be reported to give the range of p-values).

We have now modified the legends of Figures 2d and 4b-d to report on these features requested in addition to the changes related to points 1.2-1.4 below.

1.2 Fig. 2e (illustrative free energy profiles) - it is not a statistically justified interpretation to draw different profiles as different if you cannot reject the null hypothesis that the energy difference is not different than zero. Since your error bars overlap with zero (WT) for $23^{\Delta L}$ and $24^{\Delta L}$, it is not justified to draw the free energy profile for these two systems as not overlapping with the WT free energy profile. This needs to be corrected.

We agree with the reviewer's comment (with thanks) and have accordingly made changes to the figure by showing, in the schematic free energy landscape (Fig 2e, Page 7) only the double truncation ($23^{\Delta L}24^{\Delta L}$) and double insertion ($23^{+L}24^{+L}$) mutants that show a significant difference (greater than the uncertainty) in the free energy of folding compared to the WT RNC.

1.3 Fig 5a,b, the same mistake as above is made in the free energy profiles in this graph.

In the same way as above, we have edited the figure (Page 16) by showing only the insertion mutants (23^{+L} , 24^{+L}), as they are the main subject of discussion in that section and show significant differences in both binding and folding free energies compared to the WT RNC.

1.4 p-values (or stars) need to be put above each bar in the bar plots in Fig. 2e, Fig. 5a, Fig. 4d, Fig. 2d (where the statistical test is if the P_F is statistically different than wildtype), and anywhere else such data are presented in the main text or SI.

These are from NMR data, the errors of which come from either Hamiltonian Monte Carlo analysis during the fitting of both real and imaginary components of the Free induction decay (FID) simultaneously (^{19}F , Figure 2d), or spectral noise from each NMR spectrum (^{15}N , Figure 4d). For most samples we did not need biological replicates as the S/N ratio was good and the

data on several samples show high reproducibility. For instance, three biological replicates of WT+34 RNCs prepared by different researchers reveal consistent P_F of $59 \pm 5\%$, which is effectively identical to what we report in this manuscript.

2. The Methods section do not state the temperature at which the simulations were, run, this information is stated in the caption of an SI figure to be around 150 K. I assume the NMR experiments were done at room temperature (290 K). This temperature discrepancy is unexplained, and seems unjustified. The simulations should probably be rerun at the experimental temperature, or a compelling, physical reason (that I can't imagine right now) must be given as to why it is appropriate to compare simulations at 150 K to experiments at 290 K.

Again, this is a very good point. MD simulations were carried out with structure-based models and reduced units, hence the temperature of the simulations does not correspond to the physical temperature we have in the experiment. We chose the simulation temperatures (138K and 139.5K) based on the extent of folding that we observed from ^{19}F NMR measurements. The folded population of FLN5 (P_F) from these experiments are now shown in a new figure, Figure S8b. Essentially, the two simulation temperatures resulted in the extent of folding close to the experimental data at two different linker lengths ($L=31$ and 37) that monitor the significant folding transition of FLN5 on the ribosome: 138K for $L=37$ and 139.5K for $L=31$. Figure S8d shows the results for both WT and $23^{\text{AL}}24^{\text{AL}}$ RNCs at $L=31$ and 37. Although potentially due to the lack of electrostatic interactions, which were not included in these simulations, we were unable to locate a single ideal simulation temperature for recapitulating all the experimental results; by using two simulation temperatures we could not only monitor the folding transition accurately at two different linker lengths, but also observe consistent and detailed structural differences from the two different RNCs via all-atom simulations. This enabled us to capture in MD the same trend in P_F of the engineered ribosome variants from our NMR experiments. To make these points clearer we have edited the SI Methods section (page 6) for the all-atom MD simulations as follows:

Since there is no direct link between the experimental temperature and that employed in MD simulations, we undertook simulations at the two different temperatures (138K and 139.5K) found to result in the extent of folding very close to the experimental data from NMR during the FLN5 folding transition ($L=31-37$). Running at 138K also enabled isolated full-length FLN5 to remain folded, but also generated a mostly unfolded ensemble of FLN5+31 RNC and a mostly folded ensemble of FLN5+45 RNCs, consistent with the NMR observations.

3. Through the use of particular adjectives throughout the paper, there seems to be an overselling of the implications of this work. Here are a few examples:

3.1 line 75: "these loops form the basis of novel regulatory mechanisms that modulate the folding energy landscape of NCs." The fact that you perturb a molecule (the ribosome), and another molecule changes its behavior (the nascent chain) does not demonstrate a biological regulatory mechanism. To go beyond a mere physical interaction (which the authors have shown) to a regulatory mechanism, there needs to be a demonstration of an evolutionarily evolved adaptation mechanism of form or behaviour of an organism or subcellular process to changed conditions. The authors have not shown this, so the wording about regulatory mechanisms should be removed.

We agree with the reviewer's comment that 'biological regulatory mechanism' describes broader biological phenomena with more tightly and consistently governed molecular processes. Based on the reviewer's suggestion we have removed that phrase and edited the text (page 3) :

From:

"we show how these loops form the basis of novel regulatory mechanisms that modulate the folding energy landscape of NCs."

To:

"we show how these loops modulate the folding energy landscape of NCs."

This suggestion is also in line with Reviewer 1's comment, for which we removed several subjective descriptions of the implications of our work (7 corrections were made on pages 1, 3 and 20).

3.2 line 108: "shaping the trajectories of emerging NCs". Instead of shaping, a more scientific and less loaded term would be 'influencing'.

We have edited the text according to the editor's suggestion (page 3).

4. To back up the claim in Line 340, the P_F from the simulations (Fig. S8) versus the experimental P_F should be plotted against each other as another panel in Fig. S8 and a correlation and p-value reported. (This will depend on the temperature issue raised earlier, as well).

We have added Figure S8d that compares the folded FLN₅ population (P_F) at $L=31-37$ from NMR and MD simulations. The two simulation temperatures (138K and 139.5K) observe similar P_F each at $L=31$ and 37 and consistently report a higher P_F of $23^{\Delta L}24^{\Delta L}$ RNCs compared to the WT. The populations from MD do not have technical replicates, but the convergence was rigorously checked and the errors were acquired from block analysis, the depiction of which is now added in Figure S8b,c (Page 21) of the revised manuscript.

5. Line 529: "This suggests that uL23 can also relay NC folding information..." I think the phrasing of this claim is too strong given the evidence (Logic: there is different behavior of nascent chains interacting with uL23 variants, so therefore uL23 can relay information). I think a more appropriate claim (given the data) would something like "This suggests the possibility that uL23 might be capable of relaying NC folding..."

We have edited the text in the revised manuscript as per the reviewer's suggestion.

Minor:

line 134: Provide the loop lengths when referencing these: "to those observed in archaeal and eukaryotic (cytosolic) ribosomes"

We have added the length of the loops and changed the text (page 5):

From:

"bacterial ribosomes and mitochondrial ribosomes possess long tunnel loops of uL23 (~13 residues) and uL24 (~11-13 residues) relative to those observed in archaeal and eukaryotic (cytosolic) ribosomes,"

To:

"bacterial ribosomes and mitochondrial ribosomes possess long tunnel loops of uL23 (18-20 residues) and uL24 (15-18 residues) relative to those observed in archaeal and eukaryotic cytosolic ribosomes (5 residues for both uL23 and uL24),"

The equivalent of Fig. 2d needs to be shown for $L=31$. SI is fine.

This is now in Fig S3f of the revised manuscript.

Line 351: What does 'centrosymmetric' mean?

We used the term to indicate that the density of the NC in the $23^{\Delta L}24^{\Delta L}$ tunnel is symmetric with respect to the centre of the NC density due to the absence of the uL23 and uL24 loops from our MD simulations (Fig.3d). For clarity we have altered the relevant sentence (page 12):

From:

"By contrast, the NC ensemble distribution is centrosymmetric on $23^{\Delta L}24^{\Delta L}$ ribosomes where more contacts are made with H7 and H24 relative to WT RNCs."

To:

"By contrast, the NC ensemble distribution is more symmetric in relation to the centre of the tunnel on $23^{\Delta L}24^{\Delta L}$ ribosomes where more contacts are made with H7 and H24 relative to WT RNCs."

Line 562: Can you explain what the "six phases of ribosomal evolution" are? Sounds interesting.

Our mention of this is based on the accretion model (Petrov et al., <https://doi.org/10.1073/pnas.1509761112>) prokaryotic ribosomes evolved in six phases, sequentially acquiring capabilities for RNA folding, catalysis, subunit association, correlated evolution. In brief based on their explanations these phases are:

- Phase 1: ancestral RNAs form stem-loops and minihelices.
- Phase 2: the LSU catalyses the condensation of nonspecific oligomers. The SSU may have a single-stranded RNA-binding function.
- Phase 3, the subunits associate, mediated by the expansion of tRNA from a minihelix to the modern L shape. LSU and SSU evolution is independent and uncorrelated during Phase 1-3.
- Phase 4, evolution of the subunits is correlated. The ribosome is a noncoding diffusive ribozyme in which proto-mRNA and the SSU act as positioning cofactors.
- Phase 5, the ribosome expands to an energy-driven, translocating, decoding machine.
- Phase 6 marks the completion of the common core with a proteinized surface

For clarification we have changed the reference and the text (page 19) :

From:

"The exit tunnel is the sole ribosomal region that has structurally and functionally developed in all six phases of ribosomal evolution (Bowman et al., <https://doi.org/10.1021/acs.chemrev.9b00742>)."

To:

"The exit tunnel is the sole ribosomal region that has structurally and functionally developed in all six phases of ribosomal evolution *based on the accretion model*⁴⁸."

⁴⁸Petrov AS, et al. History of the ribosome and the origin of translation. *Proceedings of the National Academy of Sciences* 112, 15396-15401 (2015).

Line 48: The sentence reads strange to me by ending it with the words 'these'. Perhaps this is subjective, and the authors are free to ignore this comment if they wish.

We have now corrected the text (page 2) :

From:

"Understanding of the role of the ribosome would allow the delineation of protein folding mechanisms as they take place within cells and lead to the ability to manipulate these."

To:

"Understanding of the role of the ribosome would allow the delineation of protein folding mechanisms as they take place within cells and lead to the ability to manipulate *such processes*."

Reviewer #4

Ahn et al describe a thorough and impressive study of the co-translational protein folding process that occurs in the ribosome exit tunnel. The study applies very powerful NMR tools to monitor the folded and unfolded states of the nascent chain as well as the interactions of the chain with the exit tunnel. Using these tools, they explore the engineering of three important protein loops of the ribosome that interact with the nascent chain as it folds. They employ CRISPR/Cas9 methods to re-engineer these loops making deletions and additions, and they utilize cryoEM structures of the engineered variants combined with molecular dynamics simulations to illustrate the role that the loops within the tunnel play in folding the nascent chain. The data are quite remarkable, and the arguments are convincing.

The detailed elucidation of the L4, L22, L23, and L24 loops in assisting or manipulating folding events is quite impressive. The arguments wherein L23 and L24 work cooperatively is supported by the ability to invert the function through engineering. The combination of cryoEM structures supporting the engineered loops with the use of focused molecular dynamics, provides the detailed insight 'within' the tunnel that has otherwise been nearly impossible to discern. The correlation of the relative function of the loops and their evolution across different organisms extends fundamental understanding of protein expression and suggests opportunities for design.

We thank the reviewer for their appreciative comments

The authors seem to suggest that design of modified ribosomes may be beneficial in some biological context. This seems rather oblique as modification or re-engineering of an organism's ribosome (particularly a human) would seem to bias away from productive biology and be unmanageable for the full breadth of protein expression required of the organism. Nevertheless, the exquisite detail that these studies indicate is possible becomes a pathway for mechanistic understanding that may eventually link with sequence design to facilitate biological advantage. It will be intriguing to see if future engineering can bias the productive expression of certain fold-types.

We agree with the reviewer and our future aims in this area are indeed to explore the utility of modified ribosomes in biotechnological avenues such as recombinant expression and probe insertion to report on translation. We have added this aspect to the discussion (page 20):

Future investigations on the CTF of proteins with different fold-types (e.g., the all alpha-helical α -spectrin) will enhance the mechanistic understanding and allow exploration of the utility of the modified ribosomes in biotechnological directions, such as enhancing the expression of recombinant proteins.

Questions/More info needed:

It is not clear to what extent the particular FLN₅ CTF is selected for timescales of interaction. Will other systems be amenable or is this a special case? This may fall into categories of fold-type nascent chains, and any comments on this would be welcome.

While other proteins and folds are also observable on the ribosome e.g., alpha-synuclein (Deckert *et al.*, <https://doi.org/10.1073/pnas.1519124113>, <https://doi.org/10.1073/pnas.2103015118>), HemK and alpha-1-antitrypsin (unpublished, ongoing work in our laboratory) have been preliminarily investigated by NMR on the ribosome including the ribosomal variants in this study, FLN₅ was selected as its folding both in isolation and on the ribosome has been extensively studied by our group (Cabrita *et al.*, <https://doi.org/10.1038/nsmb.3182>, Waudby *et al.*, <https://doi.org/10.1073/pnas.1716252115>) with NMR and MD simulations. The 'special case' aspect mentioned by the reviewer is perhaps more a case of FLN being 'better established' through this underpinning understanding of the RNC system which also includes a good handle of the preparative biochemistry of these large complexes where each system requires extensive experimental establishment. However, our data with the other nascent polypeptide systems strongly suggests that these will become amenable to analogous studies. Extrapolating from our early evidence with these other systems, we expect that different sizes and fold-types of the native structure of the protein of interest will result in some variability from FLN₅ in their interactions with the ribosome and folding outcomes. Another key factor, addressed in part in our recent paper (Cassaignau *et al.* <https://doi.org/10.1038/s41557-021-00796-x>), is that the extent of NMR observability of resonances of other systems will certainly show a dependence on the specific RNC system under study. The immunoglobulin-like (Ig-like) fold is of course among the most common structural motifs found widely in proteins of diverse function. We touch upon the need for future investigations of other systems in the manuscript, stating for example that studying α -spectrin nascent chains, for example, will be useful for understanding the impact of fold-type, as it is similar in size (109 aa) to FLN₅(105 aa), but with a different fold (all-alpha). This aspect is now discussed in the new sentence added in page 20.

It is intriguing that the ¹⁹F measurements correspond to slow exchange and the ¹⁵N measurements suggest faster averaging. Can the authors be more explicit as to the timescales that the measurements are sensitive to and how this compares with the timescales suggested in the MD simulations?

As the reviewer points out, our ¹⁵N RNC spectra show averaged NMR signal arising from the ensemble of unfolded (U) FLN₅ NC conformations on the ribosome that interchange on the 'fast exchange' NMR timescale (U-U exchange, $k_{ex} \gg 10^6 \text{ s}^{-1}$) (references 4, 6, 35). These U state FLN₅ NCs coalesce into a sharp, single U state peak in ¹⁹F RNC spectra, which is in 'slow exchange' (U-F exchange, $k_{ex} \ll 380 \text{ s}^{-1}$) with the folded (F) FLN₅ NCs that give rise to a broader F state peak. The resonances of F state NCs are too broad to be visible in the ¹⁵N spectra due to NC-ribosome interactions, but the three-fold degeneracy of the ¹⁹F nucleus within its rotationally mobile CF₃ group allows the F state peak to be visible in ¹⁹F spectra. The MD simulations were run using structure-based model potentials, which operate with reduced units (the length scale, time scale, mass scale, and energy scale are all 1) and hence make a comparison of timescales from MD with that from experiments impossible. We have added a note to clarify these points on Page 6 (SI) of the revised manuscript.

There is no indication of the deposition of structural data to the Protein Data Bank. This should be done for the cryoEM structures determined.

We have deposited the data in PDB and refer to their id in SI (Page 5):

Deposition of structural data: Five cryoEM structures have been deposited to Protein Data Bank with their id - 7ZOD (23⁺), 7Z20 (23⁺L+34), 7ZP8 (WT+34), 7ZQ5 (23^ΔL24^ΔL), 7ZQ6 (23^ΔL24^ΔL+37)

The chemical shift assignments may have been previously reported. If so, then the BMRB accession number should be provided. If not, then a deposition to the BMRB of the assigned chemical shifts should be accompanied with this publication. The chemical shift assignments were previously deposited to the BMRB and the accession numbers are as follows: unfolded FLN₅ (25748), folded FLN₅ side-chain (51075), FLN₅A3A3 (51023). This information is now added in SI (Page 4).

Minor points/corrections

Page 12, upper paragraph: Suspect that the authors mean 1.8 sigma, and the font is not set to symbol.

A correction has been made accordingly (Page 12)

Page 14, last line: What is the length (in time) of the molecular dynamics simulations? The manuscript indicates 10⁸ steps, in Langevin dynamics, which does not correlate well with real time. What is the estimate of the time for comparison?

MD simulations were run using reduced units so direct comparison of the timescale is not available. This is now mentioned in the SI (Page 6).

Supplementary Information

1. Page 3, first paragraph: reference is made to Chan et al. but there is no citation information. Please update. We change this to 'Chan *et al.*, currently under final stages of revision'

2. Page 3, NMR spectroscopy: At what field strength were the ^{19}F spectra acquired?

The manuscript has been revised (SI, Page 3) to include this information (500 MHz) that was omitted in error.

3. Would like to see an example equation for computing deltaG (and delta-deltaG) from populations in NMR spectra. This would provide better confidence and understanding to readers from different backgrounds.

The manuscript has been revised to include this equation, $\Delta G_{F-U} = -RT \ln(P_F/P_U)$, in SI (Page 4)

REVIEWERS' COMMENTS

Reviewer #1 (Remarks to the Author):

My comments have been addressed, and I think its a great paper.

Reviewer #2 (Remarks to the Author):

The authors have addressed all my concerns. I accept their argument that the current manuscript represents a major effort and a significant step forward towards understanding the role of ribosome parts in nascent peptide folding. Moreover, the current experiments have been planned and executed with great care. Thus, I concur that the current work should be made available to the field now. I look forward to the results of further experiments building on the current work.

Minor suggestions (no re-review required):

1. Given that the authors in their rebuttal mention that technicalities of the CRISPR process may explain the failure to obtain uL4 mutants, I suggest changing the conclusion of the section on “Design and engineering”
 - a. unsuccessful attempts at truncating the uL4 loop may/could further suggest that modifications of the constriction site have a deleterious impact on the ribosome and cell viability, but could also be due to technical limitations of the CRISPR process. (I.e., the experiment suggests nothing, if CRISPR fails in the first place.)
2. Illustrate with a schematic sketch in Fig 1 that the secM stall sequence at the C-terminal end makes it possible to study the FLN5 domain in a stable RNC (to help the non-expert reader).
3. Mention that an secM mutant with increased efficiency is used for uL22 mutants to eliminate a potentially small effect of uL22 on stalling efficiency.

Reviewer #3 (Remarks to the Author):

The authors have not fully addressed my concerns. (I am not concerned about the Cryo-EM.)

1. I asked for the nature of the error bar to reported in the captions (i.e., whether it is a standard deviation, standard error, confidence intervals, or something else) and its procedure for being computed. The authors have not done this satisfactorily.

- Fig. 2d caption does not state what the error bar is, only how the error bar was computed.

- Fig. 2e caption doesn't state how the error was computed - presumably a propagation of error from P_F errors? Nor what the error bar represents.

- Fig. 3b caption doesn't state what the error bars are, and how they were computed.

- Fig. 4d does not state the nature of the error bars, nor do they clearly explain it in the Methods (see below), nor cite their procedure for computing them except for the statement "Errors are derived from spectral noise from each NMR spectrum (b-d)." At a minimum the nature of the error bars needs to be stated and a reference cited or a procedure explained for calculating these errors from the NMR spectrum.

- Fig. 5a, same.

In the SI Methods they state "At least two independent experiments were performed for each RNC sample, and the error was determined from the standard deviation of these experiments". The authors need to make several things clear to the reader:

1. What figures does this apply to (insert it in the caption).

2. What is the nature of the error bar from this procedure? As written this statement could be interpreted in different ways by readers: Is it the standard deviation of the two values? The standard error (i.e., standard deviation divided by the square root of $N=2$); or is it a confidence interval computed from the standard error (e.g., assuming a Gaussian distribution, the 95% Confidence Interval is $1.91 * \text{the standard error}$).

These are straightforward questions and requests. It is not a burden to report these items. After reading their response, I'm not sure why the authors did not do it after my first round of comments? I would suggest the editor give the authors one more chance (should be a 30 minute fix).

In future studies, this field needs to move forward by performing statistical tests and reporting p-values to determine whether reported differences are different.

Reviewer #4 (Remarks to the Author):

The authors have made considerable improvement in the manuscript. By addressing the range of questions raised in review, the manuscript has become clearer, more statistically grounded, and approachable by a broad audience. What was already a strong manuscript has become even better. The manuscript makes both significant contributions to the current understanding of co-translational folding and provides insights and suggestions for how to proceed to broaden this understanding. In the review process it has become clear that there are limits to the methodologies; however, the authors have a clear appreciation for these limitations and establish a pathway to broaden our knowledge and inspire other studies

REVIEWERS' COMMENTS

Reviewer #1 (Remarks to the Author):

My comments have been addressed, and I think its a great paper.
We thank the reviewer for their extensive efforts and kind words.

Reviewer #2 (Remarks to the Author):

The authors have addressed all my concerns. I accept their argument that the current manuscript represents a major effort and a significant step forward towards understanding the role of ribosome parts in nascent peptide folding. Moreover, the current experiments have been planned and executed with great care. Thus, I concur that the current work should be made available to the field now. I look forward to the results of further experiments building on the current work.

We appreciate the reviewer's insights and suggestions, which have certainly improved the paper.

Minor suggestions (no re-review required):

1. Given that the authors in their rebuttal mention that technicalities of the CRISPR process may explain the failure to obtain uL4 mutants, I suggest changing the conclusion of the section on "Design and engineering"

a. unsuccessful attempts at truncating the uL4 loop may/could further suggest that modifications of the constriction site have a deleterious impact on the ribosome and cell viability, but could also be due to technical limitations of the CRISPR process. (I.e., the experiment suggests nothing, if CRISPR fails in the first place.)

We agree with the reviewer and changed the second data & results section (Design and engineering of ribosomal mutants using CRISPR-Cas9) text (Page 5):

Along similar lines, our unsuccessful attempts at truncating the uL4 loop further suggest that modifications of the constriction site have a deleterious impact on the ribosome and cell viability[20, 33], but **unknown technical limitations of CRISPR/Cas9-based gene editing may also be the source of the aberrant behaviour of this mutant.**

2. Illustrate with a schematic sketch in Fig 1 that the secM stall sequence at the C-terminal end makes it possible to study the FLN5 domain in a stable RNC (to help the non-expert reader).

We have now added a schematic model of a stable RNC with SecM motif in Figure 1a.

3. Mention that an secM mutant with increased efficiency is used for uL22 mutants to eliminate a potentially small effect of uL22 on stalling efficiency.

We have added the following sentence in the Methods (Page 18):

AE1-SecM was used for all RNCs with uL22 loop truncations ($22^{\Delta L}$, $22^{\Delta L}23^{\Delta L}$ and $22^{\Delta L}23^{\Delta L}24^{\Delta L}$) to maintain efficient stalling, and we observed no release of the nascent chain ($\ll 2\%$ release after 12 hr) during NMR experiments.

Reviewer #3 (Remarks to the Author):

The authors have not fully addressed my concerns. (I am not concerned about the Cryo-EM.)

1. I asked for the nature of the error bar to reported in the captions (i.e., whether it is a standard deviation, standard error, confidence intervals, or something else) and its procedure for being computed. The authors have not done this satisfactorily.

We apologize to this reviewer for appearing to disregard these comments. Error bars in comparative structural biology data are often tricky to analyse for their real meaning but we have done this now as much as absolutely possible as described below for each point raised.

- Fig. 2d caption does not state what the error bar is, only how the error bar was computed. We have now added what the error bars represent: Errors are **standard deviations (s.d.)** calculated by Hamiltonian Monte Carlo analysis during the fitting of real and imaginary components of the FID simultaneously.

- Fig. 2e caption doesn't state how the error was computed - presumably a propagation of error from P_F errors? Nor what the error bar represents.

We thank the reviewer for mentioning this. We added the following in the figure legend :
Errors are calculated by propagating the errors of P_F and P_U (see Methods).

- Fig. 3b caption doesn't state what the error bars are, and how they were computed.

We added the following in the figure legend :

Errors are s.d. calculated by propagating the s.d. derived from the spectral noise of ^{13}C or ^{15}N spectra, respectively, and s.d. of nascent-chain concentrations from western blot replicates (n=3).

- Fig. 4d does not state the nature of the error bars, nor do they clearly explain it in the Methods (see below), nor cite their procedure for computing them except for the statement "Errors are derived from spectral noise from each NMR spectrum (b-d)." At a minimum the nature of the error bars needs to be stated and a reference cited or a procedure explained for calculating these errors from the NMR spectrum.

We thank the reviewer for the suggestion. We have now added the information in the legend of Figure 4d:

The changes in the free energy of binding ($\Delta\Delta G_{\text{binding}}$) in $23^{\Delta L}$, $24^{\Delta L}$ and $23^{\Delta L}24^{\Delta L}$ RNCs are calculated using the populations of unbound state NC (P_{Ub} , Supplementary Fig. 10b). Errors are s.d. and calculated by propagating the s.d. of P_{Ub} .

- Fig. 5a, same.

As the revised legends of Figs. 2e and 4d now describe (as per the reviewer's points raised above) the origin of the errors of both folding and binding free energies. We have added a

comment in the legend (Fig. 5a) to state this: **The free energies and their errors are from Fig. 2e and Supplementary Fig. 10b.**

In the SI Methods they state "At least two independent experiments were performed for each RNC sample, and the error was determined from the standard deviation of these experiments". The authors need to make several things clear to the reader:

1. What figures does this apply to (insert it in the caption).

We agree with the reviewer's pertinently raised point. We have added the descriptions of the nature and origin of the errors of each figure – the figure legends of Fig4b and 4c were also added accordingly:

(Figure 4b) **Errors are derived from spectral noise from a single measurement.**

(Figure 4c) **Errors are derived from the spectral noise for FLN5 Δ 12 and WT, while the mean and standard error from two biological repeats are shown for 23 ^{Δ} 24 ^{Δ} .**

As all figure legends have the description of the uncertainties, we have now removed the abovementioned sentence from the Methods section.

2. What is the nature of the error bar from this procedure? As written this statement could be interpreted in different ways by readers: Is it the standard deviation of the two values? The standard error (i.e., standard deviation divided by the square root of N=2); or is it a confidence interval computed from the standard error (e.g., assuming a Gaussian distribution, the 95% Confidence Interval is 1.91 * the standard error).

This is now shown in each figure legend: description of the error bar

These are straightforward questions and requests. It is not a burden to report these items. After reading their response, I'm not sure why the authors did not do it after my first round of comments? I would suggest the editor give the authors one more chance (should be a 30 minute fix).

In future studies, this field needs to move forward by performing statistical tests and reporting p-values to determine whether reported differences are different.

We appreciate the reviewer's comments and suggestions and again apologize. Without wishing to open a major debate, such analyses are often somewhat difficult to put forward with real significance when comparing structural biology data that take months of preparation for each sample but fully agree with their analysis of the situation.

Reviewer #4 (Remarks to the Author):

The authors have made considerable improvement in the manuscript. By addressing the range of questions raised in review, the manuscript has become clearer, more statistically grounded, and approachable by a broad audience. What was already a strong manuscript has become even better. The manuscript makes both significant contributions to the current understanding of co-translational folding and provides insights and suggestions for how to proceed to broaden this understanding. In the review process it has become clear that there are limits to the methodologies; however, the authors have a clear appreciation for these limitations and establish a pathway to broaden our knowledge and inspire other studies

We thank the reviewer for their supportive comments.